# Representative surface snow density on the East Antarctic Plateau

Alexander H. Weinhart[1], Johannes Freitag[1], Maria Hörhold[1], Sepp Kipfstuhl[1,2], Olaf Eisen[1,3]

[1] Alfred-Wegener-Institut Helmholtz-Zentrum für Polar- und Meeresforschung, Bremerhaven, Germany
[2] Physics of Ice, Climate and Earth, Niels Bohr Institute, University of Copenhagen, Copenhagen, Denmark
[3] Universität Bremen, Fachbereich Geowissenschaften, Bremen, Germany

*Correspondence to*: Alexander H. Weinhart (alexander.weinhart@awi.de)

**Abstract**

Surface mass balances of polar ice sheets are essential to estimate the contribution of ice sheets to sea level rise. Uncertain snow and firn densities lead to significant uncertainties in surface mass balances, especially in the interior regions of the ice
sheets, such as the East Antarctic Plateau (EAP). Robust field measurements of surface snow density are sparse and challenging due to local noise. Here, we present a snow density dataset from an overland traverse in austral summer 2016/17 on the Dronning Maud Land plateau. The sampling strategy using 1 m carbon fiber tubes covered various spatial scales, as well as a high-resolution study in a trench at 30°E 79°S. The 1 m snow density has been derived volumetrically, vertical snow profiles have been measured using a core-scale microfocus X-ray computer tomograph. With an error of less than 2%, our method
provides higher precision than other sampling devices of smaller volume. With four spatially independent snow profiles per location we reduce the local noise and derive a representative 1 m snow density with an error of the mean of less than 1.5%. Assessing sampling methods used in previous studies, we find the highest horizontal variability in density in the upper 0.3 m and therefore recommend the 1 m snow density as a robust measure of surface snow density in future studies. The average 1 m snow density across the EAP is 355 kg m$^{-3}$, which we identify as representative surface snow density between Kohnen Station
and Dome Fuji. We cannot detect a temporal trend caused by the temperature increase over the last two decades. A difference of more than 10% to the density of 320 kg m$^{-3}$ suggested by a semi-empirical firn model for the same region indicates the necessity for further calibration of surface snow density parameterizations. Our data provide a solid baseline for tuning the surface snow density parameterizations for regions with low accumulation and low temperatures like the EAP.

# 1 Introduction

Various future scenarios of a warming climate as well as current observations in ice sheet mass balance indicate a change in surface mass balance (SMB) of the Greenland and Antarctic ice sheets (IPCC, 2019). Accurate quantification of the SMB is therefore one of the most important tasks to estimate the contribution of the polar ice sheets to the global sea level rise (Lenaerts et al., 2019). Satellite altimetry is a state of the art technique to measure height changes of the major ice sheets on large spatial scales (McMillan et al., 2014; Schröder et al., 2019; Sorensen et al., 2018). These changes are converted to a respective mass gain or loss, which are directly linked to an eustatic change in sea level (Rignot et al., 2019; Shepherd et al., 2018). But this volume change converted to a mass change is subject to large uncertainties (Shepherd et al., 2012). In altimetry, at the margins of the ice sheets the local surface topography is a limiting factor in accuracy, while in the comparably flat and high-elevation interior part of the ice sheets snow properties like density have a much larger influence on the accuracy (Thomas et al., 2008). Therefore, an accurate snow and firn density on top of the ice sheets, which undergoes constantly the natural process of densification, is crucial. Given the large extent of the ice sheets, the spatial coverage of ground truth snow and firn density data is still sparse. To overcome this shortcoming, snow density in firn models is often parameterized as a function of climatic conditions, such as temperature, wind speed and accumulation rate (Agosta et al., 2019; Kaspers et al., 2004) and validated by field measurements. Then, this parameterized approach is implemented in firn models leading to a surface snow density (e.g. Ligtenberg et al., 2011). But the modeled density seems to underestimate the snow density when compared to independent ground truth data from Antarctica (Sugiyama et al., 2012; Tian et al., 2018). Inaccurate snow density, especially in the uppermost meter, leads to significant surface mass balance uncertainties (Alexander et al., 2019). Accordingly, ground truth density data are urgently needed to optimize densification models, which are crucial to convert height changes to mass changes in altimetry and therefore reduce the uncertainties in ice sheet mass balance estimates.

One source of uncertainty in the assessment of ground truth density data is the representativeness of the derived density values mainly due to the sampling strategy and sampling tools, as the snow surface on the ice sheet is spatially inhomogeneous at all scales. Apart from climate-induced (e.g. seasonal or event-based) density fluctuations, surface snow density is also influenced by topographic changes of the ice sheet surface and underlying bedrock on small (tens of meters) and large spatial scales (up to hundreds of kilometers) (Frezzotti et al., 2002; Furukawa et al., 1996; Rotschky et al., 2004). On the local scale, surface roughness and the surface slope in combination with dominant wind regimes and varying accumulation rates (Fujita et al., 2011), causes the main variations in density.

Arthern et al. (2006) derived snow accumulation in Antarctica from available field measurements of accumulation and density. To obtain this density, sampling is usually conducted in snow pits with discrete sampling over depth. Between Kohnen Station and Dome Fuji, snow density has been sampled in discrete depth intervals by Sugiyama et al. (2012), who report a high spatial variability on a kilometer scale. A small part of the variability can be attributed to the sampling method. Conger and McClung (2009) compared different snow cutting devices with various volumes between 99 cm³ and 490 cm³. The combination of under-sampling (usually negligible), variation of the device itself (0.8-6.2%) and the weight error of the scale can add up to a

significant error (dependent on the type up to 6%). Box- or tube-type cutters with larger sampling volumes are suggested for more precise measurements, with the disadvantage of coarser sampling intervals. Other commonly used devices to derive snow density in discrete intervals use dielectric properties of snow (Sihvola and Tiuri, 1986) or penetration force into the snow (Proksch et al., 2015).

In this paper, we present surface snow density data from a traverse covering over 2000 km on the East Antarctic Plateau (EAP). We show snow density data using the recently introduced liner sampling method (Schaller et al., 2016). The focus of this study is on the uppermost meter, resonating the study of Alexander et al. (2019) who emphasized the importance of an accurate 1 m density of polar snowpack. To reduce the stratigraphic noise we show a strategy with multiple samples per location. This allows a more representative local 1 m snow density. The spatial representativeness of density profiles in East Antarctica has

been recently addressed at the local scale (Laepple et al., 2016), but correlation studies for larger scales are currently not available. We discuss the representativeness of density on small and large spatial scales as well as on the temporal variability of density. Beyond improving density retrieval, our results can be of particular interest for calibration of snow density parameterizations in firn models for this part of the East Antarctic ice sheet.

## 2 Material and methods

**2.1 Study area**

We performed an overland traverse in austral summer 2016/17 – a joint venture of the Coldest Firn (CoFi) project and the Beyond EPICA – Oldest Ice Reconnaissance (OIR) pre-site survey (Karlsson et al., 2018; Van Liefferinge et al., 2018) (Fig. 1). The CoFi project aims at an improved understanding of firn densification with samples from the EAP. In its framework, five firn cores have been drilled, referred to as B51, B53 (both drilled in 2012/13), and B54, B55 and B56 (drilled on the traverse

in 2016/17).

From Kohnen Station the traverse went to former B51 drill site. Right after B51 the traverse split up and followed two different legs, to reunite at the OIR field camp at 79°S, 30°E. After accomplishing the OIR survey and drilling the firn core B54, the traverse continued to the former Plateau Station (abandoned in 1969) and then returned back to Kohnen Station.

We follow Stenni et al. (2017) using the term EAP for the region higher than 2000 m above sea level (asl).

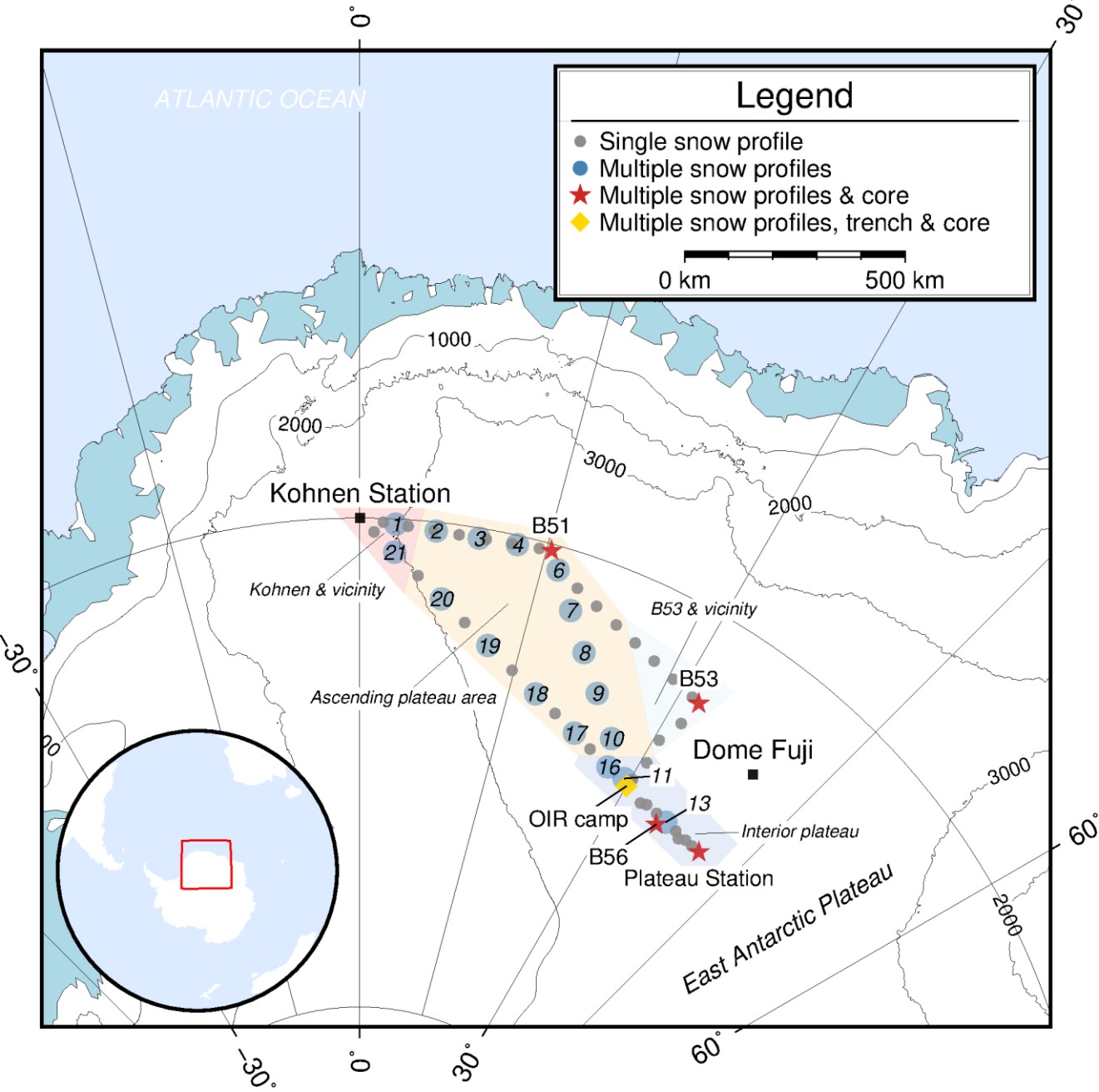

**Figure 1: Overview map of the traverse route and sampling locations, inset shows location in Antarctica. Contour lines are given in 1000 m asl intervals. The first sampling position with multiple liners after Kohnen Station is named location 1. Following the traverse route, B51 is also called location 5, OIR camp location 12, Plateau Station location 14, B56 location 15 and B53 location 22 (s. Tab. 2). 200 m firn cores were drilled at locations indicated with a red star. Subregions defined in chapter 2.5 are colored differently (Kohnen and vicinity: purple, ascending plateau area: orange, B53 and vicinity: light blue, interior plateau: lavender).**

The traverse covers a region with an annual mean temperature range of about -43°C at Kohnen Station (Medley et al., 2018; Oerter et al., 2009) to -58.4°C at Plateau Station (Kane, 1970; Picciotto et al., 1971), which belongs to the lowest firn temperatures ever recorded in 10 m depth (cf. Dome A: -58.3°C (Cunde et al., 2008)).

At Kohnen Station the accumulation rate used to be 64 kg m² a$^{-1}$ (Oerter et al., 1999) with increasing tendency to over 80 kg m² a$^{-1}$ over the last decades (Medley et al., 2018). At Dome Fuji 27.3 kg m² a$^{-1}$ was measured (Hoshina et al., 2016). For the locations along the traverse an accurate value is difficult to obtain. Large-scale accumulation estimated based on remote sensing techniques (Arthern et al., 2006) are assumed to be too high for the EAP (Anschütz et al., 2011). Karlof et al. (2005) determined an accumulation rate of ~45 kg m²a$^{-1}$ close to location 5 (Fig. 1), Anschütz et al. (2009) published ~20 kg m²a$^{-1}$ for sites between location 8 and B53 as well as OIR camp and Dome Fuji. A high inter-annual variability of accumulation rate is observed in several places on the EAP (Hoshina et al., 2016; Hoshina et al., 2014; Oerter et al., 2000). A 1 m deep snow profile can therefore cover a time period of about four years at Kohnen Station and up to 20 years on the interior plateau.

While the Northern part of the traverse (Kohnen Station – B51) is more strongly influenced by synoptic activities with periodic snowfall (Birnbaum et al., 2006), the interior plateau (OIR camp to Plateau Station) is characterized by diamond dust deposition from a clear sky atmosphere (Schwerdtfeger, 1969), which was described by Furukawa et al. (1996) as calm accumulation zone. Wind maps (Lenaerts and van den Broeke, 2012; Parish, 1988; Sanz Rodrigo et al., 2012; van Lipzig et al., 2004) show generally low mean wind speed (around 6 ms$^{-1}$) from Kohnen Station along the ice divide up the EAP, but lower values for the region around Plateau Station. Due to the prevailing Antarctic high pressure system over the EAP and the gentle slopes, the katabatic winds reach only moderate wind speeds there. While e.g. at Kohnen Station occasionally snow storms with wind speeds exceeding 15 m s$^{-1}$ can happen, this is not the case on the interior plateau.

## 2.2 Liner sampling

For clarity, we define the terms used in the following paragraphs in Table 1.

**Table 1: Definition of terms used in the following sections are listed below**

| Term | Symbol | Description |
|---|---|---|
| Liner | - | 1 m of snow sampled with a carbon fiber tube. This term is used in a methodological context or for the tube itself. |
| Snow profile | - | (Continuous) snow sample at a given position. It may consist of several consecutively (vertically on top of each other) sampled liners; the length can be 1-3 m. |
| Location | - | A given place with one or several snow profiles taken within a range of 50 m. |
| Liner density | $\rho_L$ | Volumetrically derived 1 m density of one single liner. Note: for snow profiles over 1 m length, liner densities for every meter segment are calculated individually. |
| µCT$^x$ mean density | $\rho^x_{\mu CT}$ | µCT derived mean density for the sampling interval x. |
| Location mean density | $\rho_{loc}$ | Average of liner densities at one location for the same depth interval (usually 0-1 m). |
| Horizontal standard deviation | $\sigma^x_H$ | Standard deviation of either liner density or µCT density for depth interval x over horizontal distance in a given area. Note: for 1 m we use the liner density, for smaller intervals µCT$^x$ means. |
| Vertical standard deviation | $\sigma^x_V$ | Vertical standard deviation of either µCT density over depth interval x or liner density (only for snow profiles >1 m) at a given position. |
| Standard error | $\sigma_n$ | Definition in Sect. 2.4 |

Along the traverse route, vertical snow profiles were extracted using the snow liner sampling technique, also described by Schaller et al. (2016). Each vertical profile was taken using a carbon fiber tube of one meter length and ten centimeters in diameter. The liner was pushed into the snow until the liner top was level with the snow surface. Afterwards, a snow pit next to the liner was dug and the snow was cut at the liner bottom with a metal plate to take the filled liner out of the pit wall. Both

ends were covered with a WhirlPack® plastic bag to reduce possible contamination by touching the liner ends and air ventilation. During the sampling process, the liner was handled carefully to avoid concussions that destroy the original snow stratigraphy (e.g. not to bounce against the liner with the shovel and placing it softly into the sample box). A 1 m snow profile can be retrieved within 15 minutes. The liners were stored in isolated polypropylene boxes and shipped to the Alfred Wegener Institute (AWI) in Bremerhaven in a continuous cold chain.In total 144 snow profiles in different setups and total lengths were

taken (Sect. 2.2.1 – 2.2.3). All strategies described in the following sections have been applied independently from each other.

### 2.2.1 Single snow profiles

Single profiles were taken every 30 km. On the last segment of the traverse (OIR camp to Kohnen Station) the distance increased due to limited liner availability. In total, 31 single snow profiles are available (Fig. 1).

### 2.2.2 Multiple snow profiles

22 locations with multiple profiles were sampled during overnight stops of the traverse, therefore the distance between the locations varied (roughly around 100 km). Regularly four snow profiles were sampled, at one location three, at two locations only two profiles because of time constraints (s. Tab. 2). The four profiles were arranged in an even-sided triangular setup with one profile in the center (labeled with 'X') and three profiles around it (labeled with 'A', 'B' and 'C'). The corner profiles A, B, C are on a radius of 10 m to the central profile X (Fig. 2). 83 profiles were retrieved in this setup. The locations are named

in ascending order (Fig. 1 and Tab. 2).

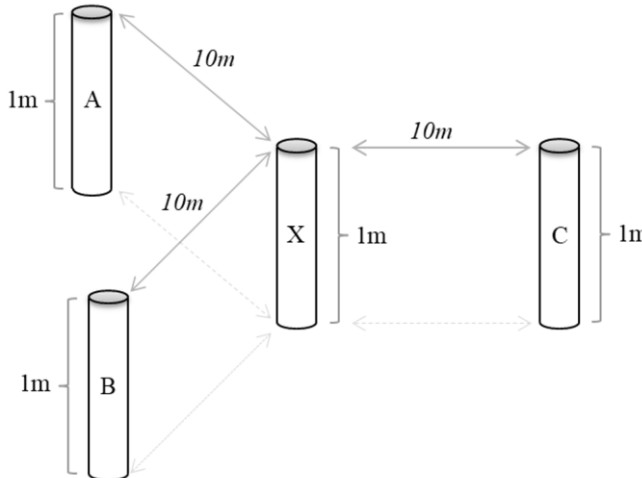

**Figure 2: The sampling setup for locations with multiple snow profiles. The profiles A, B and C have a sampling distance of roughly 10 m to the central profile. Due to time constraints, locations 19 (three profiles) 11 and 13 (two profiles) have been sampled differently.**

### 2.2.3 OIR trench

At the OIR camp (Fig. 1), a 50 meter long and ca. 2.3 meter deep trench was excavated by a PistenBully snow vehicle (Fig. 3). The trench orientation was perpendicular to the main wind direction (127° true North). Thirty 3 m snow profiles were sampled directly at the trench wall using the liner technique described above. At every sampling position in the trench three liners were taken below each other. The first liners were pushed into the snow around 0.2 meters behind the trench wall, to ensure an original stratigraphy not disturbed by excavation of the trench. After removal of the snow, the liners were directly taken out of the wall and the next consecutive liner in depth was placed at the same position (see Fig. 3, where the first liner is already in place). The lateral spacing between neighboring liners varied between 0.4 and 2.4 meters, depending on the surface structure. The profiles were taken within two days after excavation of the trench (31.Dec.2016-02.Jan.2017).

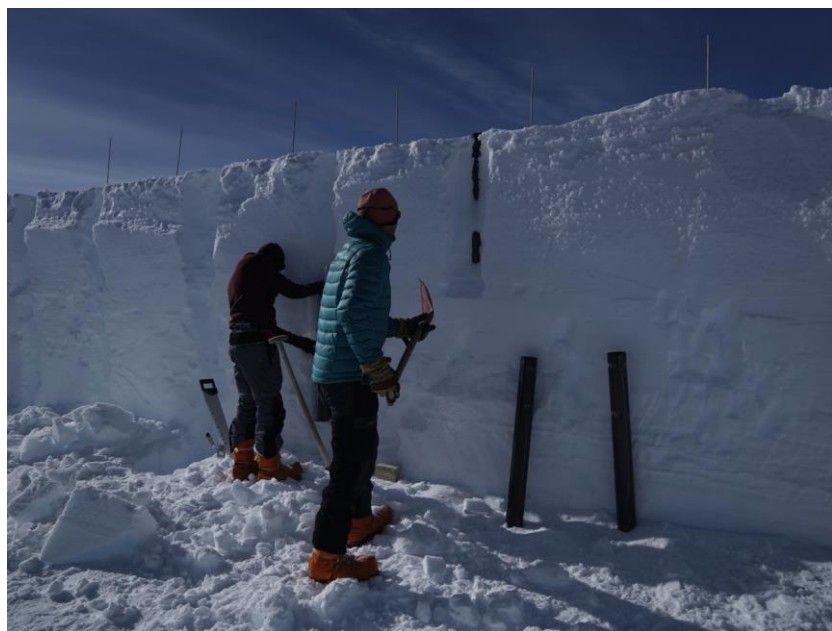

**Figure 3: Sampling procedure in the OIR trench. The first carbon fiber tube (liner) is pushed into the snow after excavation of the trench. The positions were marked with a small bamboo pole. After retrieval of the first profile, the vertically consecutive second and third liners were taken. Two empty liners lean at the trench wall. The last liner had to be dug out partly as the trench was only 2 to 2.5 meters deep.**

### 2.3 Density measurements

The snow liners have been non-destructively analyzed at AWI with the core-scale microfocus X-ray computer tomograph in a cold cell (µCT), specifically constructed for snow, firn and ice cores. For technical details see Freitag et al. (2013) and Schaller et al. (2016). Before the measurement all liners were weighed. The weight of the carbon fiber tube was subtracted. The exact height of filled snow inside the liner was determined using the µCT. Then, $\rho_L$ was calculated volumetrically. All liners have been measured in a 2D-mode using a setup of 140 kV and 470 µA at -14°C. Breaks and lost snow in the snow profiles haven been spotted during the scan and corrected (set to NaN) in the µCT density profiles, which have a vertical resolution of ca. 0.13 mm (s. Appendix).

For the calculation of the µCT density only the central segment of the liner is used as scattering effects at the outer parts of the liner occur. The used segment corresponds to less than half of the snow volume in the liner. Missing snow at the edges of the profile does not influence the µCT scan. Accuracy of the $\rho_{µCT}$ can be affected by the calibration, which is done with three cuboids of bubble free ice with different lengths in every scan individually, or at the horizontal variability on the very small-scale, as the central part of the profile can have a different density than the edges.

It is generally possible that the snow profiles are subject to compression during sampling or transport. Therefore the exact snow volume determined with the µCT is rescaled to the original 1 m length (length of every single snow profile is determined individually) to avoid this potential error source. But lost snow in the liner (or at top or bottom), e.g. in non-cohesive layers

(such as depth hoar layers), can lead to lower densities. Thus, $\rho_L$ is also affected by errors. Conger and McClung (2009) reported, that snow sampling devices with larger volumes usually result in higher precision in snow density. The volume of the snow liners (radius: 5 cm, length: 1 m) is 7855 cm³, 16 times the volume with the highest precision in their study. As the volume error among single liners is not known, we assume a 0.3 mm variation in both dimensions (length and radius), resulting in a volume error around 1.2%. As still small parts inside the liner might not be completely filled with snow (e.g. lost snow during the transport) we estimate the under-sampling error of the liner method to be less than 1.5%. Additional error sources are the precision of the used scale (1 g or 0.03% compared to the mean value along the traverse) as well as weight variations among the carbon tubes (<0.1%). The maximum relative error is estimated to be below 1.9%.

Both, $\rho^{1m}_{\mu CT}$ and $\rho_L$ are in good agreement with each other (Fig. 4), the differences between the volumetrically calculated $\rho_L$ and $\rho^{1m}_{\mu CT}$ is on average only 0.6%. As the $\mu CT$ density is sensitive to calibration, we consider $\rho_L$ to be more accurate for a 1 m interval. Some systematically higher values in the $\mu CT$ measurements can be caused by low-quality calibration in single measurements. Therefore, for the 1 m surface snow density we use $\rho_L$. For the comparison of intervals smaller than 1 m we use the $\mu CT$-derived density $\rho_{\mu CT}$ (Tab. 1).

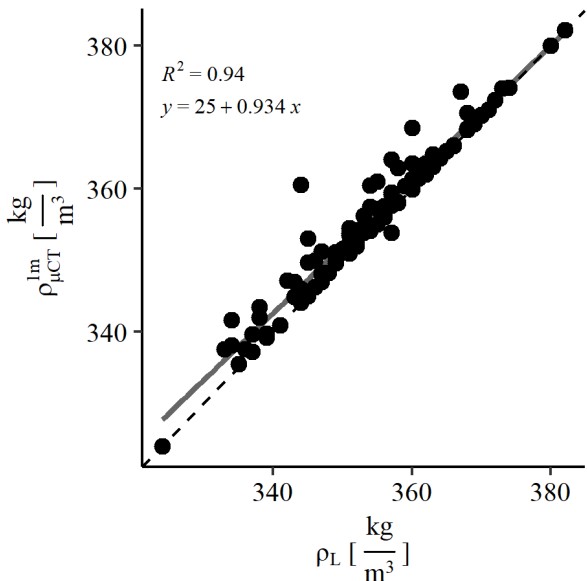

Figure 4: Comparison of liner density ($\rho_L$) with $\mu CT$ density ($\rho^{1m}_{\mu CT}$) calculated from the 114 liners along the traverse. Values of both measurements are in good agreement with an R² of 0.94. The linear fit is given with a grey solid line, the dashed black line represents x=y.

## 2.4 Finding a representative density

Fisher et al. (1985) defined stratigraphic noise as "random element caused by the surface irregularities", which is present in any taken snow profile or ice core. This stratigraphic noise is mainly caused by spatially inhomogeneous deposition in combination with wind, leading to snow patches or dune structures that usually have a spatial extent of several meters. This

stratigraphic noise hampers the representative (i.e. for a certain location or area) estimate of surface density, when not considered in the sampling strategy. To still be able to get a representative value or profile (of density or other parameters) at a given spot, several samples have to be taken at a distance, at which they are not subject to the same stratigraphic noise. For example, samples should not be taken from the same dune or snow patch, as these values cannot be considered to be spatially independent. By stacking or averaging independent samples, the stratigraphic noise is reduced. This has also been performed for e.g. isotopes (Karlöf et al., 2006; Münch et al., 2016) and a common (annual or seasonal) climatic signal can be retrieved despite a high level of stratigraphic noise.

The (minimum) sampling distance between two samples was quantitatively described for snow density by Laepple et al. (2016). In a 2D high-resolution trench study at Kohnen Station they have shown, that the correlation coefficient between single profiles decreases rapidly with increasing distance and settles at a constant value after 5-10 m. In the following we refer to samples taken at this distance as 'spatially independent'. Consequently, we consider the multiple snow profiles at one location to provide spatially independent $\rho_L$. In the OIR trench, we assume a sampling distance of 5 m between two profiles as sufficient. For a representative 1 m $\rho_{loc}$ we aim for a relative error of less than 2%. To test how many snow samples per location are needed for this representativeness, we calculated $\sigma^{1m}_H$ of $\rho_{loc}$. We used the maximum number of spatially independent $\rho_L$ for $\rho_{loc}$ (further called $n$). We did this for both the multiple liners at the traverse locations and the OIR trench. At the locations along the traverse we use all four available $\rho_L$ ($n=4$) to calculate $\sigma^{1m}_H$ of $\rho_{loc}$. In the trench, we created two sets of seven $\rho_L$ ($n=7$; maximum possible number with spatial independent samples) with different snow profiles in both sets and calculated the mean value. We then derive the standard error ($\sigma_n$), which depends on the number $n$ of $\rho_L$ at a given location by

$$\sigma_n = \frac{\sigma^{1m}_H}{\sqrt{[2;n\text{-}1]}} ,\qquad\qquad (1)$$

with the denominator being a varying number of snow profiles from 2 to n-1. This means, for example, when using seven profiles (like one set in the OIR trench) we are able to calculate the standard error for 2 to 6 profiles. In this way we use the maximum sample size without an artificially caused bias in the data. This can happen, for instance, by creating sets with small sample size and picking $\rho_L$ randomly. Accordingly by a) using large volumetric samples we improve the accuracy and by b) using several profiles at each location we improve the representativeness of the density values derived for each location. We are therefore able to deliver a more accurate and representative density of each site, compared to previous studies.

**2.5 Definition of subregions on the EAP**

We pooled several snow profiles for further investigation to characterize the surface density of a larger ($\geq$10,000 km²) region. We chose a minimum number of 10 profiles (0-1 m) per area. We followed the classification of Furukawa et al. (1996) as well as possible and used the 3500 m asl contour line as approximate boundary between different wind and accumulation regimes on the katabatic wind zone and the interior plateau (calm accumulation zone). This way we classified one major area "Ascending plateau area" (AP) with 64 profiles, covering roughly 140.000 km² between Kohnen Station and OIR camp, and the smaller "Interior plateau" (IP) with 29 profiles between OIR camp and Plateau Station (28,500 km²). We did not include

the OIR trench, as this specific location would have been overrepresented. The area around B53 (28,500 km²) was treated as a separate area as it is on the interior plateau close to the ice divide ("B53 and vicinity" – 10 profiles). Additionally, we handled the area around Kohnen Station (Ko) with roughly 10,000 km² as another separate unit ("Kohnen and vicinity" – 45 profiles). The sample availability at Kohnen Station from other studies is sufficient, several liners from other sampling programs in seasons 2015/16 (16 profiles) and 2016/17 (18 profiles) have been added to the evaluation. The areas are color-coded in the overview map (Fig. 1).

As we present density data on different scales, in this context we use the term 'local' scale for distances between profiles at one location and the area around a sampling location (i.e. tens of meters, Tab. 1). In contrast, the term 'regional' scale is used for distances between several locations (100 km to 1000 km) and areas in the dimensions of the subregions defined above. For all subsets, we present a spatial distribution of $\rho_L$ and $\rho_{loc}$.

## 2.6 Optical levelling

The relative surface elevation of the OIR trench was measured using optical levelling at each profile position and in between two consecutive profiles. Additionally, at the OIR camp and Plateau Station surface roughness transects were measured. The optical level was placed at the transect starting point. The first height measurement was done in 10 m distance to the starting point and repeated every 2 m up to 58 m distance relative to the start, resulting in 25 measuring points per transect. In total six transects have been done at one location with 1 m lateral spacing between them.

# 3 Results

## 3.1 Snow and firn density in the OIR trench

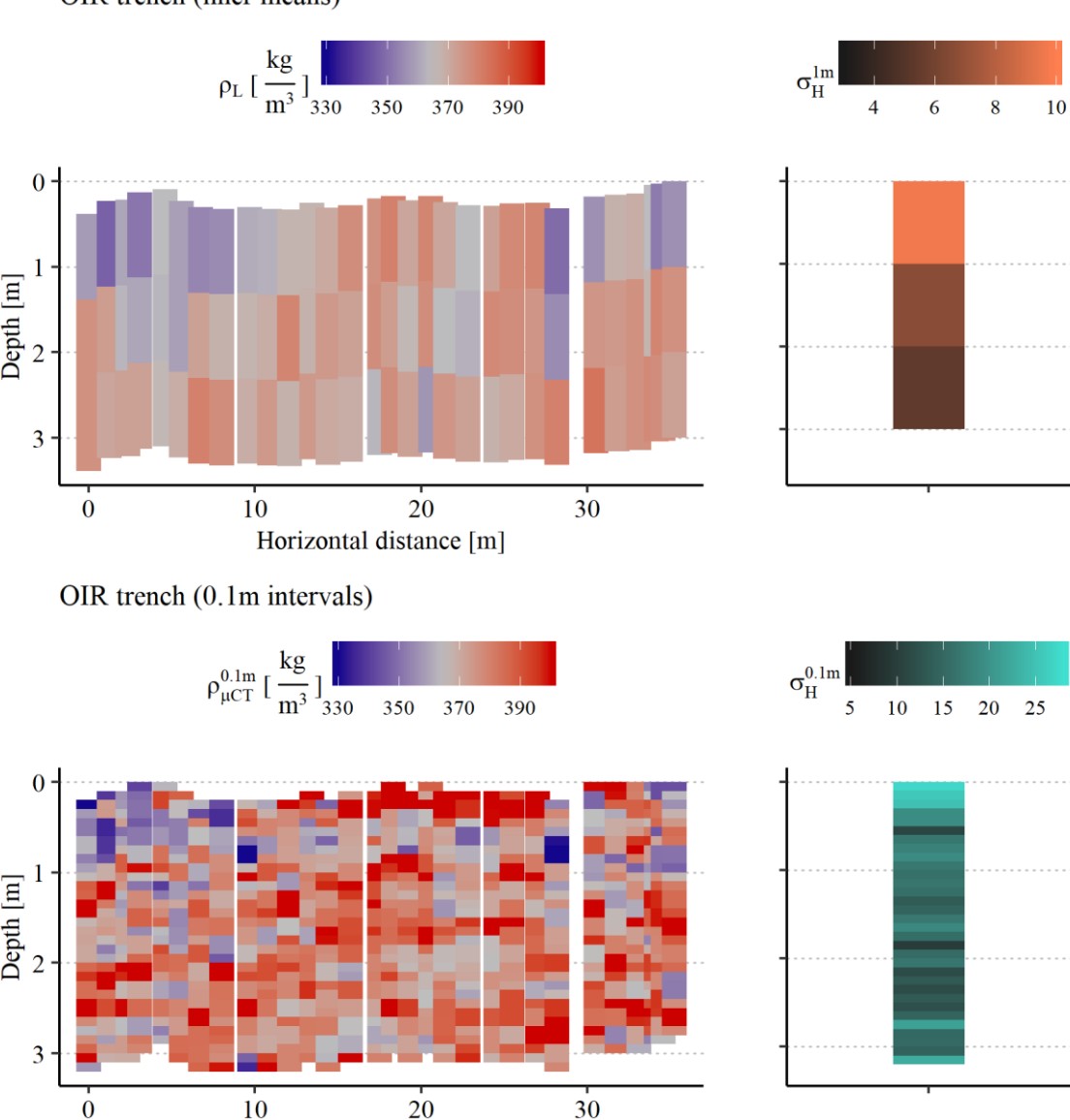

**Figure 5: Density of the OIR trench from 30 profiles in vertical 1 m (liner density, top) and 0.1 m sampling intervals (µCT density, bottom) in a color-coded plot. For the profiles in 0.1 m intervals, we used a common depth scale for the whole trench starting at the top of the profile with the highest surface elevation (profile 30), all other liners start at the measured relative height. We then calculated the density of each 0.1 m interval according to the common depth scale. $\rho_L$ and $\rho^{0.1m}_{\mu CT}$, respectively, are given in a blue (low density) to red (high density) color code. On the right of each panel $\sigma_H$ of the respective depth interval is shown.**

$\rho_L$ ranges in the OIR trench from 347 kg m$^{-3}$ to 380 kg m$^{-3}$. We calculated $\rho_{loc}$ for the OIR trench ($\pm$ standard deviation) with 365$\pm$10 kg m$^{-3}$, which is 3.1% higher than for the whole traverse (Sect. 3.2). $\sigma_H$ is between 10 and 27 kg m$^{-3}$ for 0.1 m sampling intervals and between 5 and 10 kg m$^{-3}$ for 1 m sampling intervals (Fig. 5 and Tab. 4). The highest $\sigma^{0.1m}_H$ can be found in the top 0.3 m. $\sigma^{3m}_V$ of the 3 meter profiles is 34 kg m$^{-3}$ (Tab. 4).

## 5    3.2 Snow and firn density along the traverse

Here we present data from Sect. 2.2.1 and 2.2.2. Along the traverse we find $\rho_L$ ranging from 324 kg m$^{-3}$ (pos. 22C) to 382 kg m$^{-3}$ (pos. 16A). The average $\rho_L$ calculated from 114 liners along the traverse is 354 $\pm$ 11 kg m$^{-3}$ (Fig. 6).

$\rho_{loc}$ (Tab. 1) is calculated from multiple snow profiles (Sect. 2.2.2) at each location. At location 21 and 1 close to Kohnen Station we find the lowest $\rho_{loc}$ with 344 and 345 kg m$^{-3}$, respectively. Highest $\rho_{loc}$ is found at position 5 with 372 kg m$^{-3}$
10    (Tab. 2). The average $\rho_{loc}$ along the traverse is 355 $\pm$ 8 kg m$^{-3}$. To characterize the surface variability, we calculated $\sigma^{1m}_H$ for each location separately. The minimum $\sigma^{1m}_H$ is 2 kg m$^{-3}$ at position 20 (and position 13 with only two profiles taken), the maximum $\sigma^{1m}_H$ is 15 kg m$^{-3}$ at position 22.

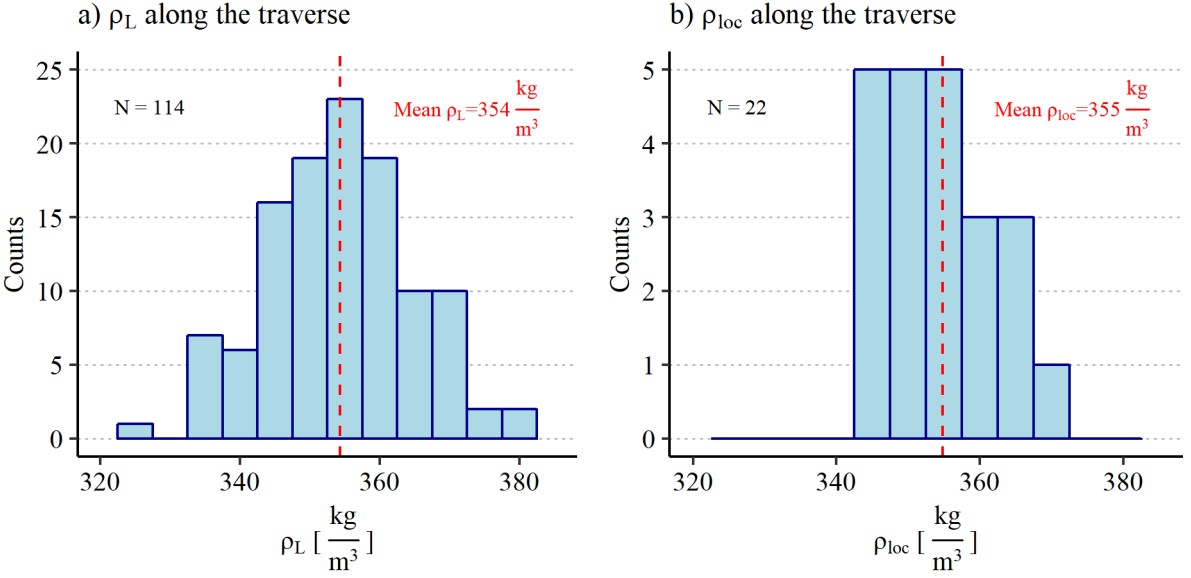

**Figure 6: Histogram of a) liner density ($\rho_L$) and b) location mean density ($\rho_{loc}$) along the whole traverse route (profiles of the OIR**
15    **trench not included). For both plots we used a bin width of 5 kg m$^{-3}$. The average liner density and location mean density, respectively, is given with the red dashed line.**

A detailed overview of all $\rho_{loc}$ and $\sigma^{1m}_H$ along the traverse can be found in table 2 and a visualization in the appendix (Fig. 13).

**Table 2: $\rho_{loc}$ at each location with multiple liners and the respective standard deviation. The number of liners at each location is given in brackets. For locations and abbreviations see Fig. 1.**

| Location (No. of $\rho_L$) | Longitude [°] | Latitude [°] | Elevation [m asl] | Sampling date | $\rho_{loc}$ [kg m$^{-3}$] | $\sigma^{1m}_H$ [kg m$^{-3}$] |
|---|---|---|---|---|---|---|
| 1 (4) | 2.89 | -75.11 | 2990 | 14.Dez.2016 | 345 | 8 |
| 2 (4) | 6.12 | -75.18 | 3146 | 15.Dez.2016 | 355 | 10 |
| 3 (4) | 9.58 | -75.21 | 3301 | 16.Dez.2016 | 360 | 13 |
| 4 (4) | 12.66 | -75.18 | 3400 | 17.Dez.2016 | 350 | 9 |
| 5 (4) – B51 | 15.4 | -75.13 | 3470 | 18.Dez.2016 | 372 | 7 |
| 6 (4) | 16.32 | -75.47 | 3484 | 19.Dez.2016 | 353 | 14 |
| 7 (4) | 18.33 | -76.19 | 3463 | 20.Dez.2016 | 346 | 8 |
| 8 (4) | 20.66 | -76.9 | 3456 | 21.Dez.2016 | 355 | 9 |
| 9 (4) | 23.19 | -77.57 | 3452 | 22.Dez.2016 | 351 | 12 |
| 10 (4) | 26.3 | -78.29 | 3455 | 23.Dez.2016 | 346 | 5 |
| 11 (2) | 29.38 | -78.89 | 3461 | 24.Dez.2016 | 350 | 6 |
| 12 (4) – OIR / B54 | 30.0 | -79 | 3473 | 26.Dez.2016 | 358 | 6 |
| 13 (2) | 35.69 | -79.18 | 3576 | 06.Jan.2017 | 362 | 2 |
| 14 (4) – B55 | 40.56 | -79.24 | 3665 | 09-11.Jan.2017 | 352 | 10 |
| 15 (4) – B56 | 34.97 | -79.33 | 3544 | 16-18.Jan.2017 | 351 | 8 |
| 16 (4) | 27.28 | -78.84 | 3416 | 23.Jan.2017 | 366 | 11 |
| 17 (4) | 22.64 | -78.5 | 3325 | 24.Jan.2017 | 358 | 7 |
| 18 (4) | 17.62 | -78.02 | 3259 | 25.Jan.2017 | 356 | 5 |
| 19 (3) | 12.03 | -77.32 | 3153 | 26.Jan.2017 | 365 | 6 |
| 20 (4) | 7.2 | -76.54 | 3067 | 27.Jan.2017 | 368 | 2 |
| 21 (4) | 2.90 | -75.67 | 2959 | 28.Jan.2017 | 344 | 7 |
| 22 (4) – B53 | 31.91 | -76.79 | 3737 | 26.Dez.2016 | 345 | 15 |
| Whole traverse (22 $\rho_{loc}$) | - | - | - | - | 355 | 8 |

### 3.3 Representativeness of surface snow density on local scales

In Fig. 7 we compare the calculated $\sigma_n$ according to section 2.4. For four spatially independent snow profiles in the OIR trench, we get a value for $\sigma_n$ of less than 1.5% (4.9 kg m$^{-3}$) relative to $\rho_{loc}$ (355 ± 2 kg m$^{-3}$). We note, that on average $\sigma_n$ in the OIR trench is higher than the average of the four areal subsets (7.0 kg m$^{-3}$ in contrast to 6.1 kg m$^{-3}$ for two profiles and 5.7 kg m$^{-3}$ in contrast to 5.0 kg m$^{-3}$ for three profiles).

Consequently, we consider four snow profiles to be sufficient for a $\rho_{loc}$ with $\sigma_n$ of less than 2%. Unfortunately, we cannot test a number of profiles higher than six. But assuming a constant $\sigma^{1m}_H$, seven spatially independent profiles are needed to assure a relative $\sigma_n$ of less than 1%.

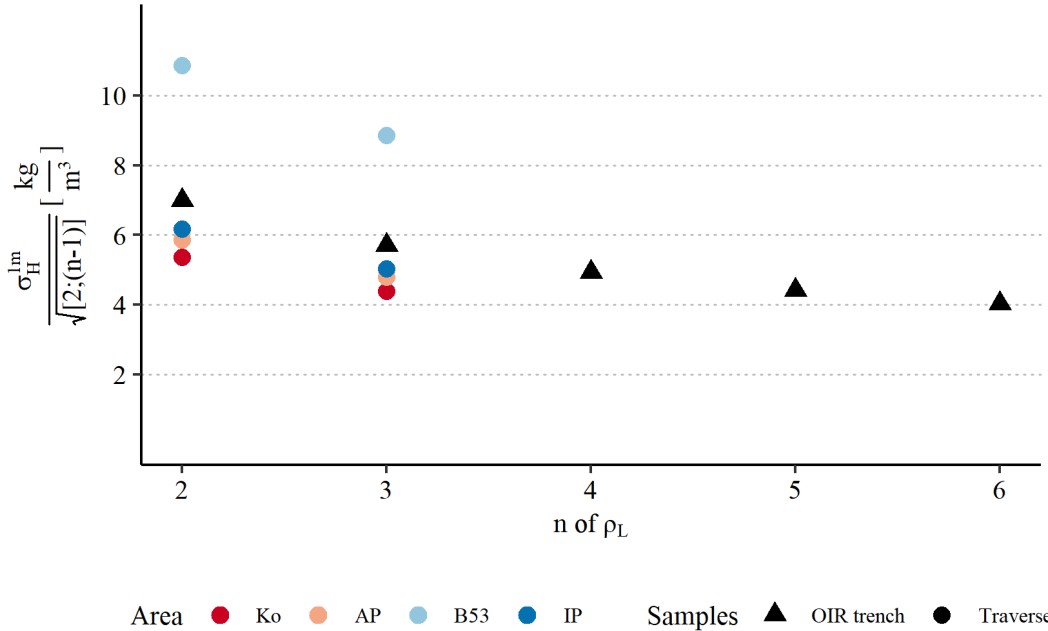

**Figure 7: Standard error ($\sigma_n$) of the location mean density ($\rho_{loc}$) as a function of the number of profiles (n). Triangles represent samples from the OIR trench while colored circles show samples along the traverse in the respective subsets (Sect. 2.5).**

### 3.4 Representativeness of surface snow density on regional scales

5   In the spatial density distribution of $\rho_L$ and $\rho_{loc}$, find similar values for Kohnen and vicinity (352±1 kg m$^{-3}$), ascending plateau area (356±1 kg m$^{-3}$) and the interior plateau (355±2 kg m$^{-3}$) (Fig. 8). These have less than 1% difference from the average value of the whole traverse. Only B53 and vicinity shows lower density values (349±3 kg m$^{-3}$, -1.7% compared to the traverse location mean density 355 kg m$^{-3}$).

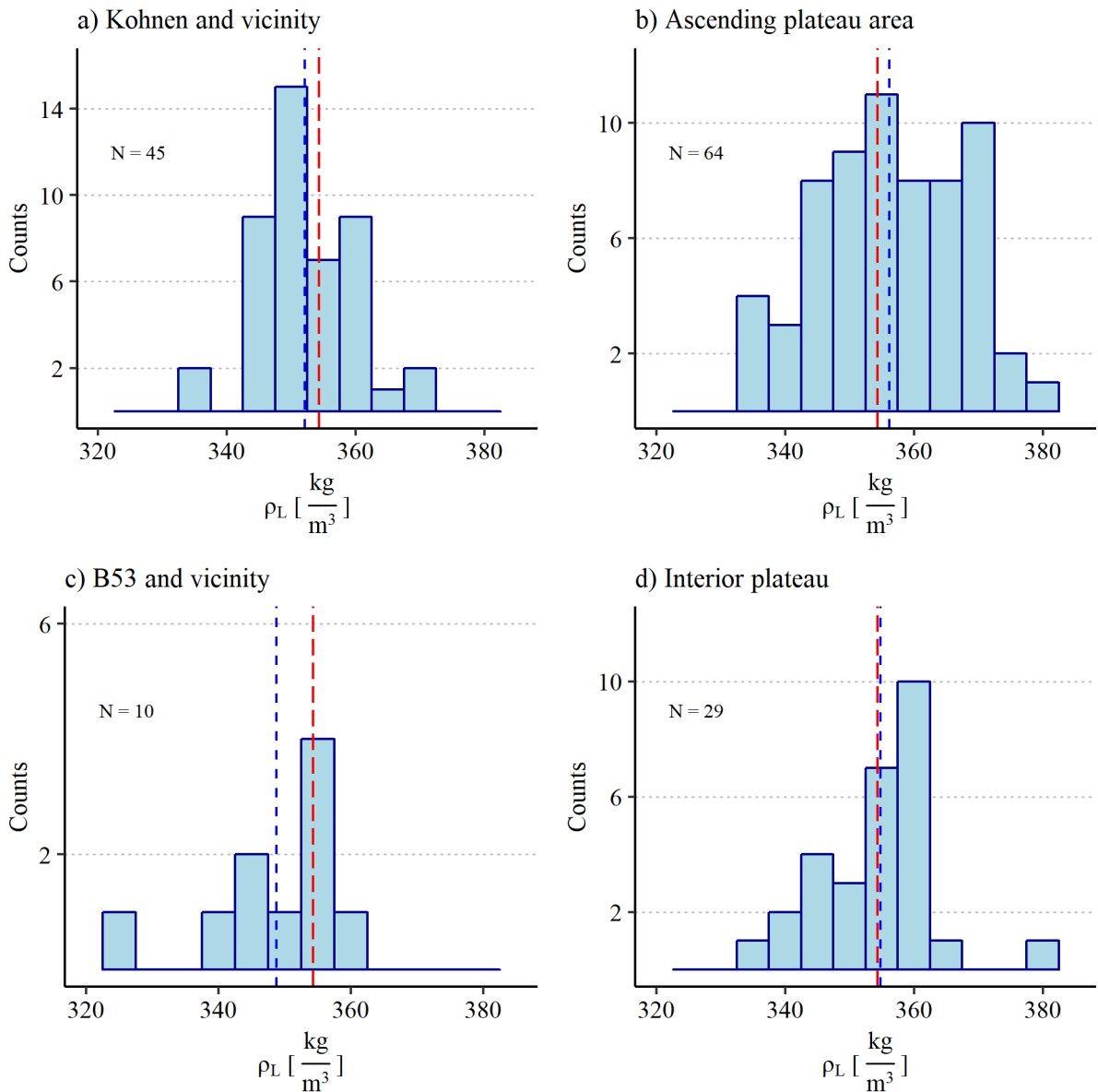

**Figure 8:** Histograms of the liner density ($\rho_L$) for the four subregions (Fig. 1). The bin width for each histogram is 5 kg m$^{-3}$. The average $\rho_L$ (Fig. 6, a) is given in a red dashed line while the liner density of the respective subregion is marked with a blue dashed line.

5    Looking at the density distribution of the high-resolution µCT density profiles (for details, see Appendix), we find a normal distribution of the snow density in the first meter (Fig. 9). We see a shift towards higher densities in the OIR trench and a higher probability for lower densities in B53 and vicinity, but in general a similar distribution of density in all subregions is found.

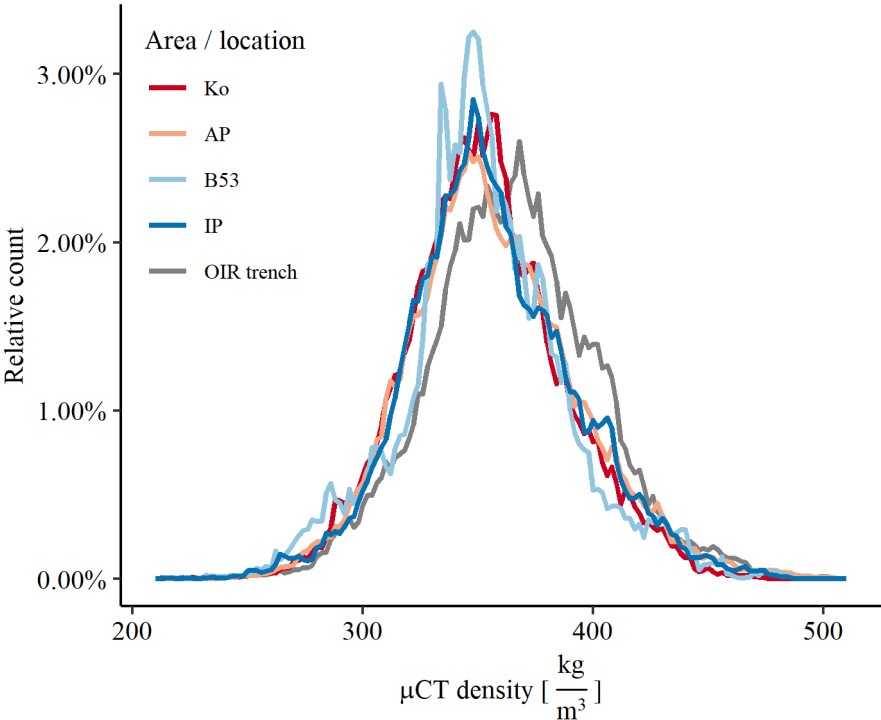

**Figure 9: Density distribution from surface to one meter depth of the µCT density. It is based on all available liners - 114 liners from the traverse (according to their subregion), 30 liners for the OIR trench (grey) and 16 liners from Kohnen Station (not this study) with a bin width of 2 kg m⁻³. We used the same color code for the subregions (Sec. 2.5) as in Fig. 1.**

5    We calculated the confidence interval (95%) of $\rho_L$ for each respective subregion (Tab. 3). We want to stress that the number of samples of "B53 and vicinity" is lower than recommended for this method. The mean value for the traverse is represented in all four intervals of the subregions. We note, that the interval for Kohnen and vicinity just includes this value.

**Table 3: Confidence intervals of 95% for each pooled area.**

| Area (number of samples) | Lower boundary [kg m⁻³] | Upper boundary [kg m⁻³] |
|---|---|---|
| Whole traverse (114) | 352 | 356 |
| Kohnen and vicinity (45) | 350 | 354 |
| Ascending plateau area (64) | 353 | 358 |
| B53 and vicinity (10) | 341 | 357 |
| Interior plateau (29) | 351 | 358 |
| OIR trench (30) | 361 | 368 |

10    The snow density directly measured at the surface in general shows high spatial variability (Figs. 5 and 10). To characterize the spatial variability of density in a given area (tens of meters for traverse locations and trenches, hundreds of meters for Kohnen Station), we use the parameter $\sigma_H$. For a comparison we used snow liners along the traverse (liners sampled at OIR trench presented in a separate column), liners from Kohnen Station (Schaller, 2018) and from East Greenland ice core project

(EGRIP) camp site (75°37′N, 35°59′W; 2702 m asl). Shown is also $\sigma_V$ for the respective areas, which can be interpreted as temporal (seasonal or annual) variations in density. We computed both ($\sigma_H$ and $\sigma_V$) for 0.1 m, 0.5 m and 1 m intervals each (Tab. 4).

**Table 4: Comparison of σ (horizontal and vertical) for each depth interval (from surface to respective depth) of samples from the traverse and OIR trench (this study), Kohnen Station and a trench from EGRIP (Schaller, 2018).**

| $\sigma^{0-X}$ [kg m$^{-3}$] | $\sigma_V$ Traverse (22 locations, 4 profiles) | $\sigma_H$ Traverse (22 locations, 4 profiles) | $\sigma_V$ OIR trench (30 profiles) | $\sigma_H$ OIR trench (30 profiles) | $\sigma_V$ Kohnen Station (16 profiles) | $\sigma_H$ Kohnen Station (16 profiles) | $\sigma_V$ EGRIP trench (22 profiles) | $\sigma_H$ EGRIP trench (22 profiles) |
|---|---|---|---|---|---|---|---|---|
| 0.1 m | 24 | 23 | 19 | 25 | 31 | 23 | 24 | 17 |
| 0.5 m | 33 | 11 | 33 | 14 | 31 | 9 | 33 | 9 |
| 1.0 m | 34 | 8 | 34 | 10 | 33 | 6 | 43 | 7 |

## 3.5 Small-scale topography at OIR camp and Plateau Station

The maximum height difference between the lowest (first) and highest (last) profile in the OIR trench is 38.5 cm. The height values of each position are given in the appendix (Tab. 6). We find significant differences in the surface topography at both places. At OIR camp the height differences between the lowest and highest point of the measured transects are 60% larger than the height differences at Plateau Station (Tab. 5). The variation of height differences between the six transects at each location is low with a standard deviation of 2.4 cm (OIR camp) and 2.0 cm (Plateau Station).

**Table 5: Maximum height differences [m] along the transects one to six at Plateau Station and B56**

| | 1 | 2 | 3 | 4 | 5 | 6 | Mean |
|---|---|---|---|---|---|---|---|
| OIR camp | 0.268 | 0.280 | 0.310 | 0.330 | 0.319 | 0.310 | 0.303 |
| Plateau Station | 0.180 | 0.211 | 0.180 | 0.174 | 0.150 | 0.212 | 0.184 |

## 4 Discussion

### 4.1 Liner method vs. discrete sampling

To discuss the 1 m snow density using the liner technique, we compare our dataset with data by Oerter (2008). In that study, snow pits with 20 km spacing have been dug and sampled along a small transect from Kohnen Station upstream towards B51 (comp. Fig. 1). A detailed map of the sampled region by Oerter (2008) is available in Huybrechts et al. (2007). Snow density has been measured volumetrically in each snow pit using discrete samples in 0.1 m depth intervals. We compare our results with density data from locations 1 to 4 (including single snow profiles in between) in two different depth resolutions (0.1 m and 1 m). For our study, we use $\rho^{0.1m}_{\mu CT}$ and $\rho_L$. For the 1 m interval from Oerter (2008) we use the average density value of all discrete samples between 0 and 1 m.

$\rho^{1m}$ from both studies are in good agreement with each other. $\rho^{1m}$ derived with the liner method tends to be 1-5% higher than the one from Oerter (2008) (Fig. 10). Higher discrepancy can be seen in the mean density of the upper 0.1 m. While we find on average $\rho^{0.1m}_{\mu CT}$=349 kg m$^{-3}$ from liner measurements, $\rho^{0.1m}$ for Oerter (2008) is 293 kg m$^{-3}$. The calculated $\sigma^{0.1m}_H$ over the

whole distance is 31 kg m$^{-3}$ for our study and 25 kg m$^{-3}$ for Oerter (2008). Interestingly, $\rho^{0.1m}$ in Oerter (2008) is always lower than $\rho^{1m}$, which is not the case in samples from our study. Due to the soft and unconsolidated snow at the surface we assume that the under-sampling error is higher at the surface for small sampling devices, which forces a systematic error towards smaller values (Fig. 10). Snow in greater depth has undergone sintering processes and is more coherent, therefore also the under-sampling error should be smaller. Additionally, a systematic error with increasing depth in the data by Oerter (2008) cannot be excluded, as the sampling device (core cutter) might densify the snow with each interval due to the thick wall in relation to the sampling volume. In contrast to other devices, the liner method preserves the original stratigraphy of the snow column. In combination with the µCT-measurement on different chosen depth intervals, this results in a density value with less uncertainty, especially for small sampling intervals at the snow surface. Despite from the sampling strategy, the difference between both datasets can be caused by different weather conditions during the sampling. This affects in particular the upper cm of the snow column.

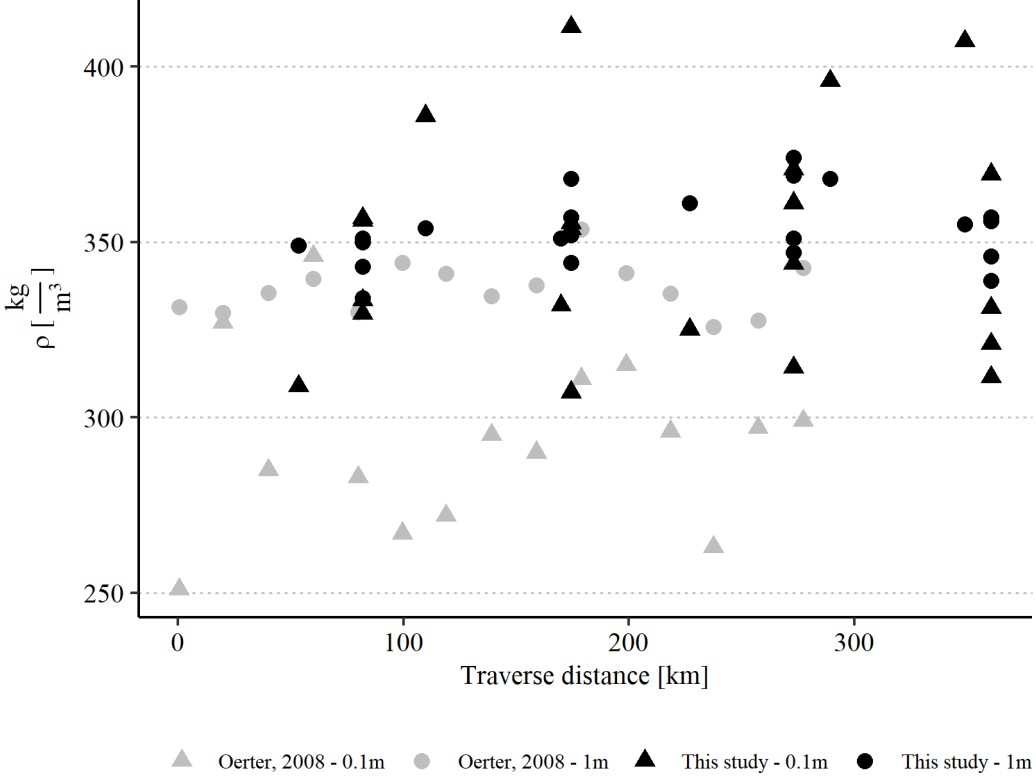

**Figure 10: Density values of this study (black) in comparison with those from snow pit sampling by Oerter (2008) (grey). The samples are taken along a comparable transect line. Density is given as mean value from the snow surface to the respective depth. The spatial variability in both, 1 m and 0.1 m intervals, can be seen by the spread of points in data of this study at one sampling location (comp. Tab. 3).**

## 4.2 Comparison of different sampling intervals

In the following we discuss the advantages of a 1 m snow density in contrast to smaller depth intervals. In this context we refer to the data presented in Tab. 4. At sites with accumulation rates higher than 100 kg m$^{-2}$ a$^{-1}$ (e.g. EGRIP), small sampling intervals (<0.5 m) do not contain the seasonal or annual variability over several years (see also data by Oerter (2008) in Fig. 10),

at sites with lower accumulation (in this context <60 kg m$^{-2}$ a$^{-1}$) the density might be masked by the high stratigraphic noise. Both effects can be seen in the low $\sigma^{0.1\,m}_V$ in contrast to $\sigma^{1m}_V$ looking at data from different sites in Tab. 4. Higher $\sigma^{1m}_V$ in snow profiles from EGRIP are caused by a clearer seasonal density cycle, which is barely or not detectable on the EAP. This can be explained with higher temperatures as well as higher accumulation rates at EGRIP. In case of surface melting like in year 2012 (Nghiem et al., 2012), $\sigma^{1m}_V$ can be even higher. We find lower $\sigma_H$ at the surface in samples from EGRIP in contrast to EAP.

This can be explained with the non-uniform deposition causing high undulations in the surface topography. We measured the topography in form of dune heights (Tab. 5), which are often 30 to 40 cm high and exceed the yearly accumulation by far. Snow layers do not form as spatially consistent as at sites where the (predicted) yearly layer thickness is larger than the amplitude of dunes. This also affects the snow density as the signal cannot form homogenously over a larger distance and causes larger $\sigma_H$. For all presented sites, the $\sigma^{0.1m}_H$ is 2.4 to 4 times higher than the $\sigma^{1m}_H$, which is explainable by the more

comprehensive density spectrum over larger depth intervals. This high horizontal variability is mainly caused by the existing small-scale topography, in particular dunes. The variability decreases below the maximum measured dune heights of 30-35 cm below the surface. These dunes have a higher snow density (Birnbaum et al., 2010) than snow that gets deposited in local depressions due to enhanced wind packing (cf. Sect 4.3).This is also visualized for the OIR trench in Fig. 5. A snow patch of low density can be seen at the surface between 0 and 5 m (horizontal distance) and rather high density between 18 and 25 m

(horizontal distance) (Fig. 5, bottom left). This illustrated well to choose a far enough distance to reduce the effect of stratigraphic noise (Sect. 2.5).

The temperature dependent densification effect does not affect the 1 m snow density substantially. By comparing all μCT density profiles over depth we cannot see a significant increase in density over the first meter. Also according to the model by Herron and Langway (1980), at a temperature of -43°C (annual mean air temperature at Kohnen Station after Medley et al.

(2018)), the increase in snow density by densification from the surface to 1 m depth is 10 kg m$^{-3}$. At a -53°C annual mean air temperature (-10°C compared to Kohnen Station) the densification is roughly 8.3 kg m$^{-3}$. A temperature change of -1°C would lower the densification induced density by about 0.17 kg m$^{-3}$.

In summary, due to the high snow density variability in the upper decimeters of the snowpack, we suggest the 1 m density as a feasible approach to derive the surface snow density independent from local recent weather conditions. For a representative

value, at least four samples should be taken per location with the respective sampling distance. The densification of snow over the first meter is negligibly small. Furthermore, we want to advert to the time efficiency of the liner method here. A 1 m snowpack density with four samples can be obtained within 1 h. Even if a high-resolution study in a snow pit is done, a snow profile using a liner can always be added to the discrete sampling in a snow pit for comparison.

## 4.3 Temporal and vertical variation of density along the traverse

Also long-term changes in temperature, accumulation rate or wind systems can affect fluctuations in density. At Kohnen Station a 1°C temperature rise per decade has been recorded by an automatic weather station, jointly operated by the Institute for Marine and Atmospheric Research (IMAU) and AWI (Reijmer and van den Broeke, 2003) over the past 20 years and discussed by Medley et al. (2018). Recent studies postulate in some areas of Antarctica, partly also on the EAP, an increase in the accumulation rate (Frieler et al., 2015; Medley and Thomas, 2019) caused by a temperature rise. However, accurate accumulation rates for the interior EAP are hard to determine and are generally overestimated (Anschütz et al., 2011).

We test the impact on surface snow density of a 1°C temperature rise as well as a 15% increase in accumulation rate at Kohnen Station. We use the parameterization after Kaspers et al. (2004):

$$\rho = 7.36 \times 10^{-2} + 1.06 \times 10^{-3}\, T + 6.69 \times 10^{-2}\, \dot{A} + 4.77 \times 10^{-3}\, W, \tag{2}$$

where $T$ is the annual mean temperature [K], $\dot{A}$ the accumulation rate [kg m² a$^{-1}$] and $W$ the mean wind speed [ms$^{-1}$]. For comparison we also use the parameterization after Sugiyama et al. (2012), as this one has been calibrated in particular with samples along a traverse over the EAP:

$$\rho = 305 + 0.629\, T + 0.150\, \dot{A} + 13.5\, W, \tag{3}$$

with $T$ in [°C], $\dot{A}$ in [kg m² a$^{-1}$] and $W$ in [ms$^{-1}$] at the given location.

A temperature rise of 1°C and an increase in accumulation rate of 15% at Kohnen Station would increase the surface snow density by 1.7 kg m$^{-3}$ according to Kaspers et al. (2004) and by 2.0 kg m$^{-3}$ according to Sugiyama et al. (2012). According to both parameterizations, the difference in density between this study and Oerter (2008) cannot be solely attributed to these climatic changes as both potential increases are inside the error range of $\rho_{loc}$. Despite of uncertainties in the precision of the sampling method or natural (climatic) variability, the discrepancy in surface density between both datasets can also be caused by stratigraphic noise over time. To give an example here, we compare $\rho_{loc}$ of snow profiles from Kohnen Station taken in two different seasons at the same position. We use 17 profiles along a transect line with 0.5 m spacing from season 16/17, which were resampled in season 18/19 (both unpublished). The climatic conditions during this time span did not change significantly. $\rho_{loc}(16/17)$ and $\rho_{loc}(18/19)$ both have the same value and the same standard deviation 350±6 kg m$^{-3}$. Although this example can give an estimate for the robustness of our density measurements using the liner method, we are not able to completely decouple the spatial variability and the temporal variability as we cannot resample the exact same position (and thus the exact same snow).

In a second test, we use an annual mean temperature of -50°C (223.15 K), accumulation rate of 40 kg m² a$^{-1}$ (0.05 m.w.eq a$^{-1}$) and a wind speed of 6 m s$^{-1}$, which are roughly the mean values of the area covered with the traverse. While the parameterization by Sugiyama et al. (2012) is fairly accurate compared to our 1 m snow density (+5 kg m$^{-3}$), keeping the temperature and accumulation rate constant we have to increase the wind speed to 9 m s$^{-1}$ to reach the surface snow density along the traverse using the parameterization by Kaspers et al. (2004).

In general we conclude, that several parameterizations for the surface snow density (Kaspers et al., 2004; Sugiyama et al., 2012) need further tuning for regions with low accumulation and low temperatures like the EAP. Rather local parameterizations should be used for regions with similar environmental conditions instead of continent-wide parameterizations.

## 4.4 A representative surface snow density on the EAP

In order to overcome the sparsity of ground truth surface snow density, regional climate models and derivatives with adequate snow deposition modules are often used to obtain estimates of accumulation and surface snow density on a full regional scale. Ligtenberg et al. (2011) presented firn density averaged from surface to 1 m depth over a period from 1979-2011. It is forced by RACMO2.3p1 mass fluxes and skin temperature and gridded at 33 km resolution.

Compared to the firn densification model presented by Ligtenberg et al. (2011), we find systematically higher values for
density on the interior EAP than the model predicts for the same locations. While $\rho_{loc}$ spans the range from 346 to 372 kg m$^{-3}$, the firn model provides a range from 308 to 332 kg m$^{-3}$ (Fig. 11). Having a sound statistics at these locations, we exclude the systematic bias to be caused by our observations, but rather assume a shortcoming of the model to yield densities which are about 10% too low. This could be caused by a multitude of reasons, e.g. model physics, spatial and temporal resolution or forcing. As the parameterization by Kaspers et al. (2004) provides density values closer to our ground truth data than the model
output by Ligtenberg et al. (2011), we suggest to revise the used slope correction (Helsen et al., 2008) for the EAP.

Our observation is consistent with recent field observations on the EAP (Sugiyama et al., 2012) or snow density collections from over two decades (Tian et al., 2018). Sugiyama et al. (2012) found a density around 350 kg m$^{-3}$ for the same depth interval (0-1 m) along a traverse between Dome F and Kohnen Station, with a similar spatial variability. Nevertheless, we cannot detect a clear trend in density along the whole traverse route. A potential reason might be the increase in elevation, distance to the
coast and major Dronning Maud Land (DML) ice divide on one hand and the decrease in temperature as well as accumulation rate (Fig. 11) on the other hand. As the sampling took six weeks in total (Tab. 2), we exclude an effect of seasonal density variability as well as a significant effect of accumulation during the traverse (as the only observed accumulation on the traverse was few diamond dust events above 3500 m asl during the nights and some drift snow). We explain the increase in surface density along the ice divide from Kohnen Station towards B51 (Figs. 8, b and 11) by smaller grain sizes due to decreasing
temperature. The combination with the lower accumulation rate and longer exposition and mixing at the snow surface seems to create a higher surface snow density here. The observation of this systematic change in density is also visible in results of Sugiyama et al. (2012) and not captured by firn models. In fact, the model by Ligtenberg et al. (2011) shows the opposite trend along this traverse section (km 0-500 in Fig. 11). High density at B51 goes along with stronger dune formation than at Kohnen Station, which was observed to increase along this traverse part, and higher potential for wind packing due to lower
accumulation rates. This is consistent with observations of dune formation at wind speeds exceeding 10 m s$^{-1}$ (Birnbaum et al., 2010) or observation of wind packing events (Sommer et al., 2018) causing increased snow density.

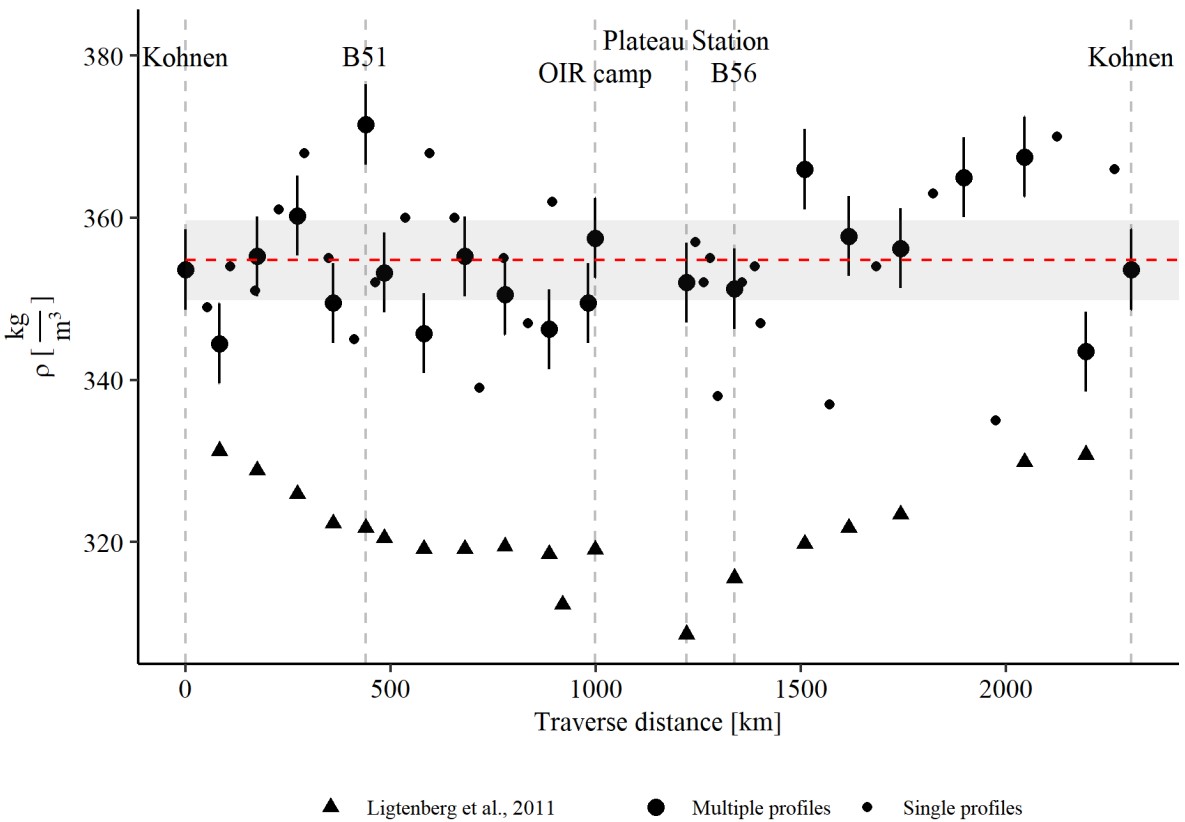

**Figure 11: Location mean density (ρ$_{loc}$) as well as liner density (ρ$_L$) along one leg of the traverse route, from Kohnen Station to B51, further along the ice divide to B53 and from Plateau Station straight back to Kohnen Station. σ$_n$ calculated from the OIR trench (Sect. 3.3) is given by vertical error bars at each location. A mean density value for Kohnen Station was calculated from samples not collected in this study (s. 2.5). The red dashed horizontal line indicates the mean density along the whole traverse, the standard error (σ$_n$) is indicated with a grey shade. The triangles show the parameterized density values according to Ligtenberg et al. (2011).**

Modelled density is parameterized by wind speed, but the process of denser packing by wind scouring and redistribution over the time until the snow is finally buried might be underestimated. We assume that the modelled low density values for the locations 14 and 15 (Plateau Station and B56, Fig. 11) in the calm accumulation zone are caused by the relatively low wind speed (Lenaerts and van den Broeke, 2012; Sanz Rodrigo et al., 2012), in combination with low temperatures and humidity (Picciotto et al., 1971). But the wind on the interior plateau is not strong enough to cause wind packing and sintering of snow crystals. It rather redistributes them smoothly at the surface, which also happens at low wind speeds. This process is significantly different from wind packing at high wind speeds. Thus the sintering process is prolonged it increases the density on the long term, which also causes an increase in density variability at the surface. But as the low densities cannot be seen for the whole interior plateau region (Fig. 8, d), we consider it rather as a process that needs very specific settings on the high plateau than an average characteristics. The abundance of wind speeds higher than 10 m s$^{-1}$ might be a limiting factor in this context.

Different environmental conditions at B53 and vicinity might cause lower density here as well (Fig. 8, c). High $\sigma_n$ for subset B53 and vicinity should not be over-interpreted, as only one sampling location with four profiles is present there. Still, $\sigma_{loc}$ is highest here amongst all locations with multiple liners along the traverse (comp. also $\sigma^{1m}_H$ in Tab. 2). An explanation can be a different wind and accumulation regime at the distant side of the ice divide causing high heterogeneity on a very small-scale.

Small fluctuations in density within the error range at nearby locations can be explained by stratigraphic noise (Laepple et al., 2016; Münch et al., 2016). Stronger variations in density, e.g. beyond one standard variation, can be caused by a complex interaction between wind speed and surface roughness on the small-scale but also have been shown to originate from dynamic interaction of ice flow over bedrock undulations, thus altering surface slope and in turn elevation and accumulation rate on the large-scale in this region (Anschütz et al., 2011; Eisen et al., 2005; Rotschky et al., 2004). For a detailed conclusion regarding

the influence of bedrock topography on the density fluctuations in our data, we consider the local scale (10 m) as too small and the regional scale (100 km) as too large. We suggest a different sampling scale (i.e. 10 km spacing of representative density) for this purpose.

As already stated above, we cannot conclusively attribute a cause to the model behavior as we also neglected the atmospheric forcing of the firn densification models, which could explain parts of the density discrepancy between field data and modeled

values. Unfortunately, it is also difficult to pin down the mechanism for the observed systematic spatial distribution of density. As the snow density parameterizations are mainly dependent on temperature and wind speed, the influence of both might be too high while processes acting on the snow surface like snow redistribution and packing play a major role on snow density. Obviously, a dedicated sensitivity study with a snow deposition and firn model is needed to discriminate the various processes affecting postdepositional snow metamorphism and densification. We suggest to set up a specific model test designed for the

EAP and use data sets like ours and those from comparable studies as the standard against which to evaluate model outcomes.

**4.5 Application to satellite altimetry of ice sheets**

Firn densification models are used in altimetry, to convert height changes of the ice sheets to mass changes. The more accurate the modelled firn density provided by these models is, the lower the uncertainties in the calculated mass changes will be. Therefore, our presented density data can be of particular interest to improve the accuracy of ice sheet mass balances.

One way in altimetry is to use a simple density mask as input parameter (e.g. McMillan et al., 2014; Schröder et al., 2019). In regions with a strong influence of ice dynamics, only the density of ice is used. In the remaining areas, also in large parts of East Antarctica where the ice flow velocities are low (Rignot et al., 2011), the density of firn is used. In this conversion, uncertainties in snow density have a direct impact on the result in mass. In our case, the 10% density underestimation in previous studies can lead to a 10% mass error (e.g. Alexander et al., 2019). Shepherd et al. (2019), in contrast, use firn or ice

density by defining areas of dynamic imbalance, which depend on surface uplift or lowering in relation to firn column changes. This method is even more sensitive to uncertainties in the firn densification models, as it subtracts variations in firn density over time.

Despite the impact of density on the height-to-mass conversion, the snowpack properties can also influence the microwave penetration into the snow and therefore considerably affect the radar altimetric measurements. Generally, snow properties like density, grain size and liquid water content can influence the permittivity (Mätzler, 1996), but also spatio-temporal variations of these parameters influence the measurements (Davis and Zwally, 1993). Furthermore layering of the snowpack seems to

affect the penetration depth, like shown in Slater et al. (2019) for Greenland. Interestingly, the density distribution of density (Fig. 9) does not show as much difference between the subregions as previously expected due to different accumulation rates. While we can see differences on the local scale (OIR trench), on the regional scale the vertical density distribution of the subregions is very congruent. Therefore further high-resolution studies on the vertical variability of the snowpack are needed on the EAP, especially with regard to high surface variability.

**4.6 Mass estimate of East Antarctica**

In the following we provide an idea how the mass of the firn column depends on the choice of the surface density using the commonly used firn densification model by Herron and Langway (1980).

Based on our findings we employ a simple quantitative calculation of the underestimated mass in the firn column with the density data presented in this study (average $\rho_{loc}$) using the semi-empirical firn densification model by Herron and Langway

(1980). We use an annual mean temperature of -50°C and an accumulation rate of 0.04 m we a$^{-1}$ as input parameters. We use the two different surface densities $\rho_0(1)$=320 kg m$^{-3}$ (Ligtenberg et al., 2011) and $\rho_0(2)$=355 kg m$^{-3}$ (this study) and sum up the water equivalent in the firn column.

We calculated 59.0 m we for $\rho_0(1)$ and 61.0 m we for $\rho_0(2)$ in the firn column down to the firn-ice-transition in 92.9 m, where scenario $\rho_0(2)$ reaches the critical density of 830 kg m$^{-3}$. The calculation is in good agreement with firn density (μCT) measured

in core B53 (unpublished data). This is roughly an underestimation in mass of 3% for the firn column only. Other effects like an overestimation of the accumulation rate on the interior plateau are not taken into account. We extrapolate this underestimation to the East Antarctic ice sheet, assuming an average ice sheet thickness of 2000 m. This results in a total underestimation of mass in the order of 1‰. Using the sea level equivalent for East Antarctica published by Rignot et al. (2019), this corresponds to about 5 cm sea level equivalent, which is more than twice as high as the sea level rise over the last

four decades (-13.9±2.0 mm, Rignot et al. (2019)). As the firn-ice-transition is not as deep at the coast as on the EAP, we consider this calculation to be somewhat overestimated.

**5 Conclusion**

We presented surface snow density data along a traverse route from Kohnen Station to former Plateau Station on the EAP using the time efficient liner method. We can reduce the sampling error from up to ±4% (Conger and McClung, 2009) by the

liner technique (this study and e.g. Schaller et al., 2016) to less than 2% relative error for a 1 m snow density. The method covers seasonal and annual variations at sites of high accumulation and reduces the influence of high surface roughness in

relation to the annual accumulation in low accumulation areas. Especially in the upper 30 cm we see the highest stratigraphic variability in snow density. As long as the accumulation does not exceed 0.5 m of snow per year (independent of the snow density), we suggest a 1 m snow density using the liner method as the best way to quantify surface snow density as the 1 m interval offers high accuracy and is representative when repeated several times. It is not biased by the seasonal density variations or weather conditions, balances high surface roughness with multiple samples, has negligible under-sampling errors as well as snow compaction and is very time efficient.

We compared the presented snow profiles to density data from snow pits by Oerter (2008). We found 1-5% lower 1 m snow densities, which cannot be attributed to a temperature change between the sampling dates only. For the density from surface to 0.1 m depth we find a considerable 16% difference in density, that we explain with a systematic sampling error. This systematic error makes comparisons of old and new datasets with different sampling devices difficult, as an increase in mass in Antarctica or an underestimation of mass in the past is hard to detect.

Especially on the EAP, field data are sparse. We conclude, that four spatially independent snow profiles are necessary to determine a snow density value with an error lower than 1.5% of the mean. To further verify this result in future studies, we suggest to test this with a similar sampling scheme with five and more profiles using the liner technique. A circular setup with one profile in the midpoint and four to six profiles along a circle with a radius of 10 m to keep spatial independency might be a feasible approach.

Our results are in good agreement with earlier density studies partly made in the same region (Sugiyama et al., 2012). We suggest a representative mean density of 355 kg m$^{-3}$ for surface snow on regional scales on the EAP. As we find a high variability on different spatial scales, we suggest to average point measurements for snow density over regional scales to find a spatially representative density value for surface snow instead of using single measurements. We divided the area covered by the traverse into subregions due to different environmental regimes, but we cannot find significant differences in surface snow density among them. Natural variability in snow density seems to be higher than previously assumed. Especially on the regional scale, we cannot see a clear correlation between temperature and accumulation rate with snow density. For future studies we therefore suggest to sample transects of 50-100 km with representative density samples every 1 km to investigate the influence of topography changes on snow density in more detail.

We also suggest further tuning of parameterizations of the surface snow density in firn models, especially for regions with environmental conditions like the EAP, which currently produce densities which are almost 10% lower than our observed values. We did not test the climatic forcing in firn models, which also can contribute to this significant offset. Neglecting the forcing, an underestimation of surface snow density can lead to a 3% mass underestimation in the firn column of East Antarctica, which roughly corresponds to a 5 cm sea level equivalent. These errors or biases in 1 m snow density can lead to large uncertainties in SMB. Improving densification models with the presented density data can also increase the accuracy of ice sheet SMB derived by altimetry, as a 10% offset in snow density, as presented in this study, can lead to a 10% error in mass. We suggest further investigation of the density variability in depth (temporal variability) with local snowpack studies in high-resolution and whether this can affect altimetry measurements.

## 6 Data availability

Datasets will be uploaded to the open-access repository Pangaea.

## 7 Author contributions and conflict of interest

JF and SK were in charge for the planning of the scientific expedition. AW and SK conducted the field work. AW performed the majority of the µCT measurements, subsequent analysis and wrote the manuscript. All authors discussed the results and contributed to revising the manuscript.

OE is Co-Editor-in-Chief of The Cryosphere.

## 8 Acknowledgements

We want to thank the whole logistic team of the traverse for technical support during the expedition and Alexandra Touzeau for assisting at taking snow liners in the field. Thanks to Melissa Mengert for conducting parts of the µCT measurements, Christoph Schaller for interesting discussions, Thomas Laepple for sharing his experience regarding representativeness of climate proxies and Ludwig Schroeder for his help regarding the impact of our data on altimetry.

The authors thank Eric Keenan, Nander Wever and Jan Lenaerts, one anonymous referee as well as handling Editor Jürg Schweizer for their sound comments.

Alexander Weinhart is funded by the German environmental foundation (Deutsche Bundesstiftung Umwelt).

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

# 10 Appendix

## 10.1 Appendix A: Snow density profile

For a better understanding of Fig. 9, we show a density profile over depth measured with the µCT. In the radioscopic image the stratification of the snowpack is visible. In Fig. 9 we took all high resolution µCT density profiles along the traverse, according to their subregion, as well as the OIR trench and plotted the relative abundance of the density values in 2 kg m$^{-3}$ intervals.

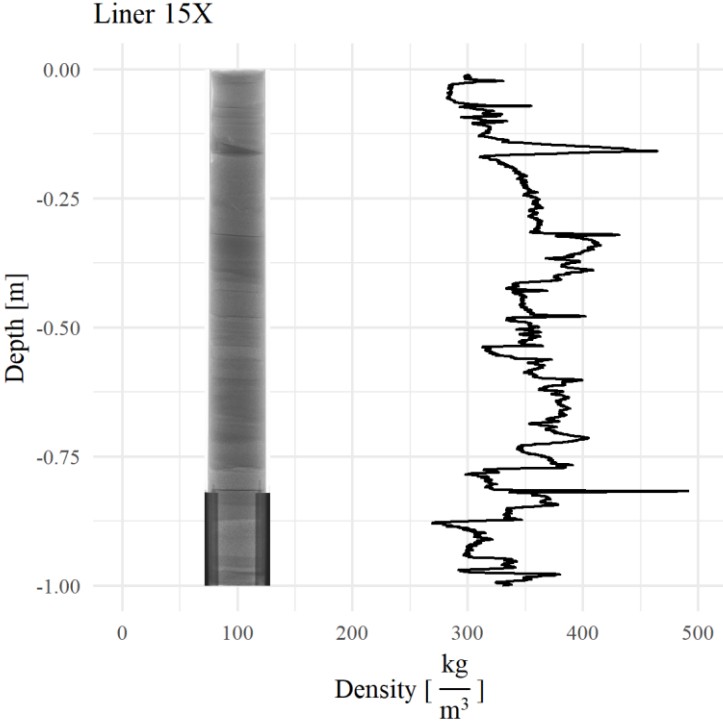

**Figure 12: µCT density of a snow profile at position 15X. On the left the radioscopic image of the snow profile is visible. Dark grey colour represents high density, bright grey represents low density values. On the right, the corresponding density profile over depth is shown.**

## 10.2 Appendix B: Geographical map of $\rho_{loc}$ and $\sigma^{1m}_H$

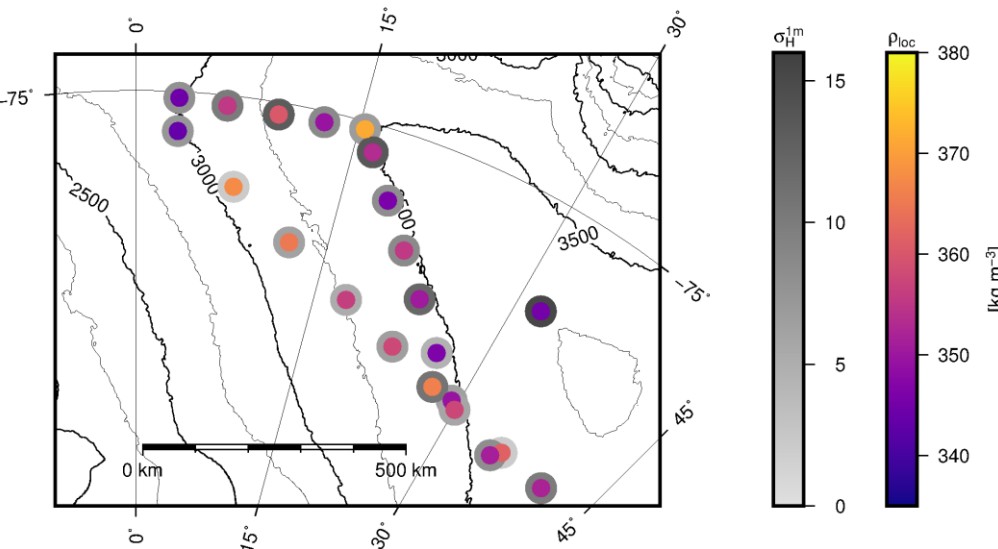

**Figure 13: Location mean density ($\rho_{loc}$) and the horizontal standard deviation ($\sigma^{1m}_H$) along the traverse. The according values can be found in table 2. Colored points show $\rho_{loc}$, grey edges $\sigma^{1m}_H$.**

## 10.3 Appendix C: Height measurements along the OIR trench surface

**Table 6: Surface levelling along the OIR trench. Surface height was measured at and in between subsequent sampling positions. In column two we show the distance along the trench, in column three the relative surface height in relation to the last profile.**

| Sample position | Distance [cm] | Relative surface height (to profile 30) [cm] |
|---|---|---|
| 1 | 0 | -38.5 |
|   | 59 | -31 |
| 2 | 125 | -23.4 |
|   | 178 | -33 |
| 3 | 237 | -21.6 |
|   | 274 | -16.1 |
| 4 | 309 | -12.7 |
|   | 391 | -17.8 |
| 5 | 462 | -9.7 |
|   | 510 | -19.3 |
| 6 | 556 | -22.7 |
|   | 610 | -30.1 |
| 7 | 672 | -30.2 |
|   | 740 | -34.7 |
| 8 | 800 | -32.1 |
|   | 895 | -35.7 |
| 9 | 970 | -30.5 |
|   | 1030 | -32.6 |

| | | |
|---|---|---|
| 10 | 1088 | -32.3 |
| | 1150 | -35 |
| 11 | 1209 | -33.4 |
| | 1278 | -33.5 |
| 12 | 1343 | -24.9 |
| | 1395 | -30.4 |
| 13 | 1440 | -31.3 |
| | 1510 | -29.6 |
| 14 | 1575 | -28.4 |
| | 1675 | -30.4 |
| 15 | 1750 | -20.2 |
| | 1790 | -19.6 |
| 16 | 1832 | -17.6 |
| | 1880 | -20.4 |
| 17 | 1934 | -22.6 |
| | 1998 | -25.2 |
| 18 | 2056 | -17.1 |
| | 2100 | -25 |
| 19 | 2145 | -24.6 |
| | 2230 | -27.9 |
| 20 | 2282 | -28.2 |
| | 2380 | -30 |
| 21 | 2449 | -28.9 |
| | 2500 | -26.8 |
| 22 | 2545 | -25.6 |
| | 2619 | -27.8 |
| 23 | 2700 | -25.2 |
| | 2760 | -29.9 |
| 24 | 2815 | -31.8 |
| | 2940 | -30.9 |
| 25 | 3051 | -18.3 |
| | 3120 | -21.9 |
| 26 | 3177 | -16 |
| | 3245 | -12.7 |
| 27 | 3310 | -14.6 |
| | 3368 | -8.7 |
| 28 | 3412 | -4.6 |
| | 3432 | -4.2 |
| 29 | 3453 | -3.2 |
| | 3488 | -1.1 |
| 30 | 3522 | 0 |