# Peer review of "Representative surface snow density on the East Antarctic Plateau"

_The Cryosphere, 2020_

## Referee Comment (RC1) · Anonymous Referee #1 · 25 Mar 2020

Review of

Representative surface snow density on the East Antarctic Plateau

by Alexander H. Weinhart

General

This paper describes density observations along two over-ice transects from Kohnen station to Dome F on the plateau of the East Antarctic ice sheet. The observational techniques are state-of-the-art, resulting in small errors and highly significant results. These results show that 0-1 m average density shows little variation along the traverse, with a mean value of about 355 kg m-2. This is an important result, as it can be used to improve the snow/firn modules in (regional) climate models and the interpretation of

satellite altimetry observations. However, the writing needs to be improved, as many formulations are unclear (for some examples, see below, but this listing is not exhaustive). The figure quality can also be improved in places.

Major comments

p. 1, l. 25: "The difference in the total mass equivalent of measured and modelled density yields a 3% underestimation by models, which translates into 5 cm sea level equivalent. " It is unclear how these numbers are obtained, see comment below on Section 4.3.

p.2, l.3: " Accurate quantification of the current state and rate of change of SMB is therefore one of the most important quantities..." A quantification is not a quantity. Please critically re-assess your formulations to improve this throughout the paper.

p. 3, l. 5: " The coldest 10 m firn temperature is recorded at Plateau Station (...), which makes the area the best modern analog of glacial firn." This is another example of a sentence that is really hard to understand. Coldest on Earth? What do you mean by "an analogue of glacial firn"? Please clarify.

Section 4.3: It is unclear to me how the density errors in previous studies lead to the SMB error results in a 5 cm sea level equivalent? Over what period? SMB is usually derived from regional climate models that quantify mass directly, i.e. irrespective of density. Mass changes by GRACE are also direct mass measurements, only satellite altimetry suffers from uncertainties in the density of the material at which the elevation change takes place, but this is valid for changes in elevation, not for steady densities as presented here.

Section 4.4: Are the Ligtenberg (2011) data also valid for the first 1 m? I think they use a simple parametrization to calculate surface density, hardly a 'model' as it is called here. It would also be valuable to provide the time span covered by the first 1 m of snow, this will vary with accumulation. In how far can climate variability be responsible

for part of the differences with other studies?

Based on its findings, does the current paper recommend to redefine 'surface density' as the average density of the first m? If so, this is an important recommendation that could be made more explicit.

Minor comments

p. 1, l. 10: "Wrong estimates of snow and firn density can lead to significant underestimations of the surface mass balance." underestimations of -> uncertainties in

p. 1, l.17: "liner" ? This has not been explained yet, so please don't use it here.

p. 1, l.23: "provided by a regional climate model" These models usually don't 'provide' density, but either prescribe it or use a simplified expression based on temperature, wind etc. Suggest to replace by 'used'.

p. 1, l. 25 and further: Note that regional climate models DO explicitly calculate accumulated mass, so using a wrong surface density does not influence the surface mass balance directly, only indirectly (through blowing snow threshold friction velocity, vertical heat transport in snow affecting surface temperature and hence sublimation etc.).

p. 2,l.2: on -> of; Greenlandic -> Greenland.

p. 2, l. 5: " Satellite altimetry is state of the art" -> Satellite altimetry is a state of the art method/technique...

p.2, l.13: " snow density is parameterized " -> snow density in models is often parameterized

p. 2, l. 30: This sentence is unclear, please reformulate.

p.3, l. 4: Remove " In order to avoid misunderstandings"

p. 3, l. 5: coldest/warmest temperatures -> lowest/highest temperatures (change this throughout the text, please)

[Figure]

p.9, l. 8: good -> well

Figures: Please include solid axes where missing.

p. 18, l. 23: This AWS has been installed and serviced by Utrecht University and AWI; please provide proper credit.

---

## Referee Comment (RC2) · Anonymous Referee #2 · 16 Apr 2020

The authors present a suite of highly accurate surface density measurements taken during a traverse in Dronning Maud Land, East Antarctica. These observations have the potential to offer the ice sheet scientific community a unique and very useful dataset to evaluate and improve snow and firn densification models. The authors present principally interesting spatial analysis of the measured snow density, which adds to the presented data. That said, we have significant concerns that should be addressed before publication, namely 1) a more detailed description of density uncertainty quantification, 2) the method used to quantify the impact of density on surface mass balance retrieval, and 3) a more detailed description of observed small scale density variability in the top 1 m presented in Figure 5, as well as its potential drivers and implications for interpretation of satellite altimetry observations. Please find a more detailed descrip-

tion of these suggestions and others broken into major and minor comments below.

Reviewed by: Eric Keenan, Nander Wever, Jan Lenaerts

Major Comments

Section 2.1: This section would benefit from a general discussion of weather and climate conditions in the area, in relation to how they may impact surface density (in terms of variability of yearly accumulation rates, wind speed and temperature). This would help setting the stage for the discussion in section 4.4.

P7,L12: "Breaks and lost snow in the snow profiles haven been corrected." This needs more explanation.

P8,L9-11: "It is generally possible that at the liner top and bottom some snow is lost, but as the exact snow volume is determined with the $\mu$CT, we overcome this error source." It's not clear how the microCT can compensate for errors due to lost snow. It's the same liner that's measured by microCT and the scale, so if the snow is lost, both methods should be affected.

P8, L15: "Therefore…." I don't see how this sentence follows logically from the previous section.

P8, L19: "spatially independent" Not really clear. Is the measurement setup in Fig. 3 considered spatially independent? I.e., are liners X, A, B, C considered spatially independent? According to the text they are, but those four liners are not really independent.

Section 2.4: This section is difficult to comprehend, and is written very compact. Particularly, please expand on: "This way we use the maximum sample size without an artificially caused bias in the data."

Figure 5: The large variability in observed density, particularly in the top meter, is very interesting and is a very nice inclusion in this paper. Can you please elaborate on what

might cause this (surface topography? winds?) and what this variability means for interpretation of satellite altimetry. In particular, the observed variability is in apparent contrast with the title of this paper "Representative surface snow density...". If surface density is highly variable, is a representative density truly the best approach or should the scientific community make an effort to model this variability? A related comment is that Section 3.5 is really short and only mentions results. It's not clear which conclusions the authors draw from this and how it is important to understand surface density variability.

P11,L1-3 and in the following sections: If the standard deviation for 1m sampling intervals is 5-10 kg/m3, how can the error quantification for the average be only +/- 2 kg/m3?

Figure 8: Observations report a near uniform mean density in the four different subregions. For me, this is a surprising result. How might the different dates the observations were taken affect the measured density, i.e. do you expect a seasonal cycle in surface density. The way this dataset is currently presented, does not take into account this possibility. Additionally, if you are not already planning on doing so, can you please include exact observations date and time in the final dataset publication on Pangaea?

P18,L18: This line mentions natural variability due to antecedent weather conditions. Section 4.2 needs to put the analysis based on climatological trends in perspective to possible year-to-year variability due to antecedent weather conditions. Since accumulation depths in dunes could be up to 30cm, this may impact top 1m density significantly.

Section 4.3: The authors aim to provide the impact of density on the uncertainty in SMB, but they fail to do this correctly. First of all, 3% uncertainty in the firn column does not directly translate to a 3% uncertainty in the overall firn+ice column (since there is much more ice than firn on East Antarctica). Secondly, this calculation pertains to mass, not mass balance (i.e. the change in mass per unit time). Instead, the authors

should think about focusing on the application to altimetry, which needs surface density to convert volume to mass. As most of the elevation changes on East Antarctica measured by altimetry are SMB-driven, the observed elevation change will be associated to the layer of recently accumulated/ablated snow/firn, with an extremely spatially and temporally variable density. Since this (near-)surface density is much more variable than at 1 m, this volume-to-mass conversion is highly uncertain, especially when focusing on small scales such as in this study. The error here is directly proportional to density, i.e. there will be 100% error in mass change if the assumed density is 100% different than it is in reality.

Figure 11: How is surface roughness and sub-grid topography, e.g. using REMA, related to observed density in this figure?

It is recommended to analyze the liners from Kohnen station from different seasons (as mentioned P9,L15/16) to show to what extent there is year-to-year variability.

P21,L5-8: First of all, a primary source of error in modeled snow density by the Ligtenberg et al. (2011) model could as well result from the meteorological driving data for the FDM simulations. Second, the text now seems to imply that more snow redistribution leads to lower densities. However, it has been demonstrated that snow redistribution tends to increase hardness/density (see Sommer at al. 2017, 2018).

We thank the authors for taking the substantial time and effort to collect, describe, and distribute these density observations. That said, we believe these observations would best serve the community if they were also included in a unified and publicly available dataset such as SUMup (Montgomery et. al., 2018).

Minor Comments P1, L11: Underestimations or overestimations.

P1, L25: Density errors can be due to errors in parameterizations or atmospheric forcing.

P2,L3: "Greenland Ice Sheet"

P2,L3-5: Please reformulate. Either "accurate quantification is important" or "The current state and rate are some of the most important quantities . . ."

P2,L11: "Especially in the interior of the ice sheets, the exact surface snow density is a limiting factor in precision." Please amend why that is, with appropriate references if available.

P2,L23: "Small variability" –> "A small part of variability" I assume.

P3,L1 and L9: "stratigraphic noise" please explain.

P3,L3-6: "In this paper,we present surface snow density data with high precision from a traverse covering over 2000km on the East Antarctic Plateau (EAP). In order to avoid misunderstandings we follow Stenni et al. (2017) using the term EAP for the region higher than 2000m above sea level (asl). The coldest 10m firn temperature is recorded at Plateau Station (Picciotto et al., 51971), which makes the area the best modern analog of glacial firn." Don't understand this section. Please explain in more detail.

P3, L10 Average local snowpack density.

P4, Fig 1: Please add elevation contour labels.

P6, L20: The sentence "The trench surface was measured. . . " needs to be placed before P6,L19: "The total height difference between the lowest . . ."

P7, L9 weighted → weighed?

P8, Fig 4: What explains the occasional large difference? Please add linear regression statistics.

P8,L15: "Therefore, to quantify the 1m snowpack density we use L, to investigate smaller intervals we use the $1m\mu CT$(Tab.1)" Since $1m\mu CT$ is CT density over 1m, and L the liner density over 1m, how should it be interpreted that $1m\mu CT$ can assist in investigating smaller intervals? Or should it read $0.1m\mu CT$.

P8, L17 Snow density profiles?

P9, Section 2.6 Optical leveling needs to be placed before Section 2.2.3., since the optical leveling is already mentioned there.

P9,L17: "Furthermore, a possible effect of the station itself should not be migrated into the other subsets." Please explain what effects are meant here.

P10, Figure description: What is meant by raster?

P12, Table 2: Can you create maps of p_loc and sigma_1m?

P18, L9: Please quantify dune height

P19, L4: What exactly leads you to make this claim. Could density errors be due to errors in atmospheric forcing? Temporal variability in snow density?

Many figures have missing axes. Please correct.

References: please provide doi's for easy lookup of literature.

References

Montgomery, L., Koenig, L., and Alexander, P.: The SUMup dataset: compiled measurements of surface mass balance components over ice sheets and sea ice with analysis over Greenland, Earth System Science Data, 10, 1959–1985, https://doi.org/10.5194/essd-10-1959-2018, https://www.earth-syst-sci-data.net/10/1959/2018/, 2018

Sommer, C. G., Lehning, M., & Fierz, C. (2017). Wind tunnel experiments: saltation is necessary for wind-packing. J. Glaciol., 63 (242), 950–958. doi: 10.1017/85jog.2017.5386

Sommer, C. G., Wever, N., Fierz, C., & Lehning, M. (2018). Investigation of a wind-packing event in Queen Maud Land, Antarctica. Cryosphere, 12(9), 2923–2939. doi: 10.5194/tc-12-2923-201

---

## Author Comment (AC1) · 25 May 2020

**Interactive comments on**
**"Representative surface snow density on the East Antarctic Plateau" by Alexander H. Weinhart et al.**

5    *Comments by the referees will be displayed in italics,* the response from the authors in normal font.

**1 Review by Anonymous Referee #1**

*This paper describes density observations along two over-ice transects from Kohnen station to Dome F on the plateau of the East Antarctic ice sheet. The observational techniques are state-of-the-art, resulting in small errors and highly significant results. These results show that 0-1 m average density shows little variation along the traverse, with a mean value of about*

10   *$355 \ kg \ m^{-3}$. This is an important result, as it can be used to improve the snow/firn modules in (regional) climate models and the interpretation of satellite altimetry observations. However, the writing needs to be improved, as many formulations are unclear (for some examples, see below, but this listing is not exhaustive). The figure quality can also be improved in places.*

We thank the anonymous referee for his feedback on our manuscript. We carefully will go through the manuscript again and try to rewrite unclear passages and elaborate further on the influence of the presented data on satellite

15   altimetry.

Generally we will include axes in the figures where missing.

**1.1 Major comments**

*p. 1, l. 25: "The difference in the total mass equivalent of measured and modelled density yields a 3% underestimation by models, which translates into 5 cm sea level equivalent." It is unclear how these numbers are obtained, see comment below on*

20   *Section 4.3.*

See comment on section 4.3.

*p. 2, l. 3: "Accurate quantification of the current state and rate of change of SMB is therefore one of the most important quantities..." A quantification is not a quantity. Please critically re-assess your formulations to improve this throughout the paper.*

25   Thank you very much for this advice. We will clarify unclear wording or passages in the text, including the mentioned one. We noticed also necessary improvements especially in section 2.4 and 4.3 (s. below).

*p. 3, l. 5: "The coldest 10 m firn temperature is recorded at Plateau Station (...), which makes the area the best modern analog of glacial firn." This is another example of a sentence that is really hard to understand. Coldest on Earth? What do you mean by "an analogue of glacial firn"? Please clarify.*

30    We do not have access to glacial-climate firn but firnification during glacial climate periods is modelled to calculate for example the $\Delta$age (the gas age-ice age difference), and to infer the phase relationship between temperature derived from the isotopes and the $CO_2$ concentration measured in ice cores. Modelling glacial-climate firn faces some problems, e.g. at the pore close off (firn to ice transition). Firnification models simulate a deeper pore close off than $\delta^{15}N$ data predict. In this sense we understand modern firn from the coldest regions of the EAP as the best modern

35    analogue of glacial-climate firn (for some regions, e.g. Kohnen station). The acronym CoFi stands for Coldest Firn. Within the project, five 200 m firn cores have been drilled on the EAP to investigate the firn densification.

In general, we will move the project description to section 2.1 and add climatic information about the area. The snow profiles presented here were taken in the framework of this project. But as this information is not necessary for the further manuscript, we probably decide to remove this sentence.

40    *Section 4.3: It is unclear to me how the density errors in previous studies lead to the SMB error results in a 5 cm sea level equivalent? Over what period? SMB is usually derived from regional climate models that quantify mass directly, i.e. irrespective of density. Mass changes by GRACE are also direct mass measurements, only satellite altimetry suffers from uncertainties in the density of the material at which the elevation change takes place, but this is valid for changes in elevation, not for steady densities as presented here.*

45    This comment also refers to your general comment on a proper use of terms, here we should rather refer to simply mass instead of a mass balance.

We want show the underestimation in mass in the firn column running the model by Herron and Langway (1980) with two different initial densities. According to our calculation, this underestimation (3% of the firn column) in East Antarctica can lead to an underestimation in total mass of 5 cm sea level equivalent.

50    We will update the section with proper terms and describe our line of thoughts at length. We also add another section with focus on the impact of our findings on satellite altimetry of ice sheets. For this we will discuss our data in more detail interdisciplinary with colleagues working with remote sensing data.

*Section 4.4: Are the Ligtenberg (2011) data also valid for the first 1 m? I think they use a simple parametrization to calculate surface density, hardly a 'model' as it is called here. It would also be valuable to provide the time span covered by the first*

55    *1 m of snow, this will vary with accumulation. In how far can climate variability be responsible for part of the differences with other studies? Based on its findings, does the current paper recommend to redefine 'surface density' as the average density of the first m? If so, this is an important recommendation that could be made more explicit.*

The suggestion to redefine the term surface snow density is an important point that will be addressed. Also in preparation of this manuscript we stumbled upon several opinions about how 'surface snow density' is defined ('fresh snow density', 'near surface density' and 'density of the uppermost snow layer' are used sometimes arbitrary for different purposes). Generally, for density smaller intervals than 1 m a lot of additionally different parameters would have to be considered. For practical reasons, 1 m intervals are not only a commonly used interval for density, but also stable water isotopes in surface snow (e.g. Masson-Delmotte et al., 2008). According to the dataset description, the data from Ligtenberg et al. (2011) are modelled (IMAU Firn Density Model) for the near-surface (0-1 m depth) and gridded to 33 km resolution.

The problem of time span and the – in turn – advantage of 1 m density compared to smaller intervals, is tackled in the conclusion, but should be emphasised in the discussion as well. As mentioned before, we added a section about the climatic setting of the area. This includes information about accumulation rates as well as temperature. At Kohnen station the accumulation rate is 64 mm we a$^{-1}$ (we = water equivalent) (Oerter et al., 1999) with increasing tendency over the last decades (Medley et al., 2018), at Dome Fuji 27.3 mm we a$^{-1}$ has been measured (Hoshina et al., 2016). For the locations along the traverse a precise value is difficult to obtain. A 1 m snow profile therefore can cover a time period of four years at Kohnen station to 20 years on the remote Plateau.

Regarding your comment on climate variability: if we understood correctly, you ask whether changing temperature can be responsible for a difference between two density datasets. Here we want to refer to section 4.2, in which we elaborate on this thought. We show, that climatic driven changes in density are too low to ascribe the difference in density between the datasets only to temperature.

**1.2 Minor comments**

*p. 1, l. 10: "Wrong estimates of snow and firn density can lead to significant underestimations of the surface mass balance." underestimations of → uncertainties in*

*p. 1, l. 17: "liner"? This has not been explained yet, so please don't use it here.*

*p. 1, l. 23: "provided by a regional climate model" These models usually don't 'provide' density, but either prescribe it or use a simplified expression based on temperature, wind etc. Suggest to replace by 'used'.*

*p. 1, l. 25 and further: Note that regional climate models DO explicitly calculate accumulated mass, so using a wrong surface density does not influence the surface mass balance directly, only indirectly (through blowing snow threshold friction velocity, vertical heat transport in snow affecting surface temperature and hence sublimation etc.).*

*p. 2, l. 2: on → of; Greenlandic → Greenland.*

*p. 2, l. 5: "Satellite altimetry is state of the art" → Satellite altimetry is a state of the art method/technique...*

*p. 2, l. 13: "snow density is parameterized" → snow density in models is often parameterized*

*p. 2, l. 30: This sentence is unclear, please reformulate.*

*p. 3, l. 4: Remove "In order to avoid misunderstandings"*

*p. 3, l. 5: coldest/warmest temperatures → lowest/highest temperatures (change this throughout the text, please)*

*p. 9, l. 8: good → well*

*p. 18, l. 23: This AWS has been installed and serviced by Utrecht University and AWI; please provide proper credit.*

*Figures: Please include solid axes where missing.*

95      All minor comments are taken as suggested by the referee.

Herron, M. M. and Langway, C. C.: Firn Densification - an Empirical-Model, Journal of Glaciology, 25, 373-385, doi: 10.3189/S0022143000015239, 1980.

100   Hoshina, Y., Fujita, K., Iizuka, Y., and Motoyama, H.: Inconsistent relationships between major ions and water stable isotopes in Antarctic snow under different accumulation environments, Polar Science, 10, 1-10, doi: 10.1016/j.polar.2015.12.003, 2016.

Ligtenberg, S. R. M., Helsen, M. M., and van den Broeke, M. R.: An improved semi-empirical model for the densification of Antarctic firn, Cryosphere, 5, 809-819, doi: 10.5194/tc-5-809-2011, 2011.

Masson-Delmotte, V., Hou, S., Ekaykin, A., Jouzel, J., Aristarain, A., Bernardo, R. T., Bromwich, D., Cattani, O., Delmotte,
105   M., Falourd, S., Frezzotti, M., Gallee, H., Genoni, L., Isaksson, E., Landais, A., Helsen, M. M., Hoffmann, G., Lopez, J., Morgan, V., Motoyama, H., Noone, D., Oerter, H., Petit, J. R., Royer, A., Uemura, R., Schmidt, G. A., Schlosser, E., Simoes, J. C., Steig, E. J., Stenni, B., Stievenard, M., van den Broeke, M. R., de Wal, R. S. W. V., de Berg, W. J. V., Vimeux, F., and White, J. W. C.: A review of Antarctic surface snow isotopic composition: Observations, atmospheric circulation, and isotopic modeling, Journal of Climate, 21, 3359-3387, doi: 10.1175/2007jcli2139.1, 2008.

110   Medley, B., McConnell, J. R., Neumann, T. A., Reijmer, C. H., Chellman, N., Sigl, M., and Kipfstuhl, S.: Temperature and Snowfall in Western Queen Maud Land Increasing Faster Than Climate Model Projections, Geophysical Research Letters, 45, 1472-1480, doi: 10.1002/2017gl075992, 2018.

Oerter, H., Graf, W., Wilhelms, F., Minikin, A., and Miller, H.: Accumulation studies on Amundsenisen, Dronning Maud Land, Antarctica, by means of tritium, dielectric profiling and stable-isotope measurements: first results from the 1995-96 and
115   1996-97 field seasons, Ann Glaciol, 29, 1-9, doi: 10.3189/172756499781820914, 1999.

---

## Author Comment (AC2) · 25 May 2020

**Interactive comments on**
**"Representative surface snow density on the East Antarctic Plateau" by Alexander H. Weinhart et al.**

5   *Comments by the referees will be displayed in italics,* the response from the authors in normal font

**2 Review by Eric Keenan, Nander Wever, Jan Lenaerts**

*The authors present a suite of highly accurate surface density measurements taken during a traverse in Dronning Maud Land, East Antarctica. These observations have the potential to offer the ice sheet scientific community a unique and very useful dataset to evaluate and improve snow and firn densification models. The authors present principally interesting spatial*

10   *analysis of the measured snow density, which adds to the presented data. That said, we have significant concerns that should be addressed before publication, namely*

*1) a more detailed description of density uncertainty quantification,*

*2) the method used to quantify the impact of density on surface mass balance retrieval, and*

*3) a more detailed description of observed small scale density variability in the top 1 m presented in Figure 5, as well as its*

15   *potential drivers and implications for interpretation of satellite altimetry observations.*

*Please find a more detailed description of these suggestions and others broken into major and minor comments below.*

    We kindly thank Eric Keenan, Nander Wever and Jan Lenaerts for their detailed and productive feedback. The deliberate comments will definitely help to improve this manuscript, in particular the discussion and application of the presented dataset.

20     Regarding the impact on satellite altimetry, we will consider adding another section in the manuscript.

**2.1 Major comments**

*Section 2.1: This section would benefit from a general discussion of weather and climate conditions in the area, in relation to how they may impact surface density (in terms of variability of yearly accumulation rates, wind speed and temperature). This would help setting the stage for the discussion in section 4.4.*

25     We agree and will rearrange the section and add a climatic overview about the area (especially temperature, accumulation rates). This should also clarify unclear passages in the text like (see also minor comment on P3, L3-6).

*P7, L12: "Breaks and lost snow in the snow profiles haven been corrected." This needs more explanation.*

s. response below

*P8, L9-11: "It is generally possible that at the liner top and bottom some snow is lost, but as the exact snow volume is determined with the μCT, we overcome this error source." It's not clear how the microCT can compensate for errors due to lost snow. It's the same liner that's measured by microCT and the scale, so if the snow is lost, both methods should be affected.*

s. response below

*P8, L15: "Therefore…" I don't see how this sentence follows logically from the previous section.*

We realized that parts of section 2.3 were not coherent enough and will update it. Then the logical sequence should be clearer.

Regarding the missing snow: You have to distinguish between lost snow at the liner top or bottom and lost snow in the middle of the profile, we also will make it clear in the manuscript. The length of every individual snow profile is measured within the procedure of the μCT scan, we do not assume 1 m as standard length for every profile (e.g.: 1 cm lost snow at the bottom equals a 99 cm profile). This way we overcome an under-sampling error due to lost snow at top or bottom. In contrast: some mm lost snow at the edge of a depth hoar layer affects the under-sampling error of the volumetrically derived density. But this does not affect the μCT density, as only a central segment is used for the density profile reconstruction and – if a break is continuous over the whole horizontal plain – positions with breaks are spotted and values set to NaN.

Still, for 1 m segments, the volumetrically derived density has a higher precision as Fig. 7 shows; we will explain in more detail the text.

*P8, L19: "spatially independent" Not really clear. Is the measurement setup in Fig.3 considered spatially independent? I.e., are liners X, A, B, C considered spatially independent? According to the text they are, but those four liners are not really independent.*

Fisher et al. (1985) defined local noise as "random element caused by the surface irregularities", which is present in any taken snow profile or ice core. Stratigraphic noise and depositional noise are used as equivalent terms for local noise as they are more descriptive. This noise is mainly caused by spatially inhomogeneous deposition in combination with wind, leading to snow patches or dune structures that usually have a spatial extent of several meters.

To still be able to get a representative value or profile (of density or other parameters) at a given spot, several samples have to be taken at a distance, at which they have not recorded the same depositional (or stratigraphic) noise. For example, samples should not be taken from the same dune or snow patch. By stacking or averaging the samples, the noise is minimized. The (minimum) sampling distance between two samples was quantified by Laepple et al. (2016) with 5-10 m, as the correlation factor between single profiles decreases with increasing distance. Also the sampling distance in this study was chosen according to this finding. In an (unpublished) test using our density profiles we also

come to a similar result. Note: Laepple et al. (2016) sampled perpendicular to the dune direction like we did in the OIR trench. For sampling parallel to the wind, the sampling distance between two samples should be higher.

This condition (not recording the same depositional noise several times) was called 'spatially independent' in our manuscript. The problem of stratigraphic noise is generally higher in regions with low accumulation and has also been recorded for e.g. isotopes (Karlöf et al., 2006; Münch et al., 2016).

For a better understanding, we will add passages of the explanation above in the text.

*Section 2.4: This section is difficult to comprehend, and is written very compact. Particularly, please expand on: "This way we use the maximum sample size without an artificially caused bias in the data."*

For the whole section, there might be many different approaches to determine a certain number of precision for our data. The method we use has also recently been applied in a study by Dallmayr et al. (in review). We are aware of the not-straightforward method and will try to explain in more detail. With 'artificially caused bias' we mean the instance of arbitrary picking sets of different numbers of $\rho_L$ (e.g. a certain number of independent profiles out of the 30 trench profiles). Instead we suggest to take the maximum possible number from the beginning. For more clarity, we also put the formula

$$\sigma_H^{1m} \Big/ \sqrt{x}$$

in a more explicit position in the manuscript.

*Figure 5: The large variability in observed density, particularly in the top meter, is very interesting and is a very nice inclusion in this paper. Can you please elaborate on what might cause this (surface topography? winds?) and what this variability means for interpretation of satellite altimetry. In particular, the observed variability is in apparent contrast with the title of this paper "Representative surface snow density...". If surface density is highly variable, is a representative density truly the best approach or should the scientific community make an effort to model this variability? A related comment is that Section 3.5 is really short and only mentions results. It's not clear which conclusions the authors draw from this and how it is important to understand surface density variability.*

The surface topography is definitely one of the driving factors for the high horizontal variability of density we see in the top 20 cm in the OIR trench. It can be seen as a complex interaction between accumulation (a combination of calm diamond dust deposition as well as event-based accumulation), redistribution of soft snow by moderate wind and the existing topography, with the topography being the dominant factor (height & shape). Additionally, long residence time of snow at the surface due to low accumulation rates enhances the chance of metamorphism and sublimation (due to the vertical temperature gradients and low humidity), which also can have an impact on the surface snow density. We will expand the discussion of the driving factors in section 4.1.

As we mention in sect. 2.2.3 and the previous comment, the OIR trench was excavated perpendicular to the main wind direction, which causes a higher variability than parallel to the wind. We will include this in the discussion.

Indeed, the high (local) variability at the snow surface (especially in the upper 20 cm) is a major finding of this manuscript. But as a consequence this is also the reason, why we argue for the use of a 1 m mean density as a more robust parameter for surface snow density. This way the density variability at the uppermost surface is compensated by using enough depth (or "annual layers", in other terms) without having the influence of a densification effect. We will emphasize this statement in the conclusions.

Otherwise, the high variability at the surface can also be an argument for a representative density obtained with the method we presented. Especially as altimetry measurements have a certain footprint area, representative regional density values can be of particular interest.

Despite the argument for the 1 m snow density, from our point of view a representative vertical variability is a more important aspect than modelling the horizontal variability. This is why we included the density distribution of density (Figure 9) in the manuscript and calculated the distribution also according to the presented subareas. The vertical variability can be interpreted as a measure of snowpack layering (i.e. layers of high and low density). The layering definitely has a strong influence on the densification from snow to ice on one hand (and therefore also on the bubble close-off at the firn-ice transition and subsequent effects). On the other hand, also satellite altimetry can use this information of vertical layering due to penetration depth into the snow and reflection at layer boundaries. We will elaborate to what extent we can include this topic in this manuscript, as the topic of snowpack layering demands another dimension (esp. further parameters to describe the snowpack) that goes beyond the current manuscript frame.

Regarding section 3.5: We included this – indeed – very short section as a baseline for the argument of dune height and surface topography on the East Antarctic Plateau. We mention the relation of dune height to accumulation rate later in the manuscript. One might argue, that the dune height of 30-40 cm is not that high. But in contrast to the annual accumulation of ~10 cm, the dune height becomes enormous. We will mark in the text when we refer to the measured surface heights in the text.

*P11, L1-3 and in the following sections: If the standard deviation for 1m sampling intervals is 5-10 kg/m3, how can the error quantification for the average be only +/- 2 kg/m3?*

We assume, you refer to the horizontal standard deviation, as also displayed in Fig. 5.

The horizontal standard deviation ($\sigma^{1m}_H$) for the liner mean density ($\rho_L$) in the OIR trench is 9.63 kg m$^{-3}$. The standard error then is defined as the standard deviation divided by the square root of the samples size: ($\sigma^{1m}_H / \sqrt{30}$) = 1.76 kg m$^{-3}$ ≈ 2 kg m$^{-3}$

120 *Figure 8: Observations report a near uniform mean density in the four different subregions. For me, this is a surprising result. How might the different dates the observations were taken affect the measured density, i.e. do you expect a seasonal cycle in surface density. The way this dataset is currently presented, does not take into account this possibility. Additionally, if you are not already planning on doing so, can you please include exact observations date and time in the final dataset publication on Pangaea?*

To be honest, we also expected to see a larger difference between the subregions (in both, mean density (Fig. 8) and density distribution (Fig. 9)) and a clearer trend along the traverse before the measurements.

To clarify your remark we add the sampling date at each location in table 2 (we planned to add the sampling date to the Pangaea dataset). From first (14.12.2016) to last (28.01.2017) sampling date we do not expect to see a seasonal cycle or bias due to sampling, especially as we did not notice significant accumulation during the traverse (main synoptic features: some diamond dust above 3500 mSL during the nights, during some days moving/drifting snow. The temperatures varied between -20 and -40°C during the night). We also add a sentence on that to section 4.4.

From extensive sampling programs at Kohnen station we can generally say, that it is possible to detect seasonal cycles in density profiles – but only with a sufficient amount of samples (if local noise can be eliminated) (Laepple et al., 2016). On the EAP, this may be more difficult as the accumulation rate is much lower. To derive a representative density profile from the OIR trench can be a part of a future study.

*P18, L18: This line mentions natural variability due to antecedent weather conditions. Section 4.2 needs to put the analysis based on climatological trends in perspective to possible year-to-year variability due to antecedent weather conditions. Since accumulation depths in dunes could be up to 30cm, this may impact top 1m density significantly.*

In this context, high mean density of single profiles can also be explained with a high percentage of dune snow in a profile. A major problem here is that we do not know anything about the persistence of the surface roughness or surface features. If the surface is reshaped once or twice a year by stronger wind (>10 ms$^{-1}$), it is hard to attribute this variability to a variability of climate. Also the distribution of snow accumulation during the year is poorly known (keeping in mind the differences at Kohnen station with higher synoptic influence than the remote plateau with diamond dust deposition).

We will elaborate on the year-to-year variability a bit further by comparing samples from Kohnen station from different seasons (15/16, 16/17, 18/19, maybe more) and present values in the next version of the manuscript. Here we want to show the high influence of noise in contrast to climatic factors on density.

For further discussion about the climatic influence on surface snow density, we want to refer to section 4.2., where we test, whether the discrepancy in density between two datasets can be ascribed to rising temperatures in DML. We come to the conclusion, that the warming of 1°C alone cannot explain the difference in 1 m surface snow density and ascribe this to the term 'natural variability' or 'noise'.

*Section 4.3: The authors aim to provide the impact of density on the uncertainty in SMB, but they fail to do this correctly. First of all, 3% uncertainty in the firn column does not directly translate to a 3% uncertainty in the overall firn+ice column (since there is much more ice than firn on East Antarctica). Secondly, this calculation pertains to mass, not mass balance (i.e. the change in mass per unit time). Instead, the authors should think about focusing on the application to altimetry, which needs surface density to convert volume to mass. As most of the elevation changes on East Antarctica measured by altimetry are SMB-driven, the observed elevation change will be associated to the layer of recently accumulated/ablated snow/firn, with an extremely spatially and temporally variable density. Since this (near-)surface density is much more variable than at 1 m, this volume-to-mass conversion is highly uncertain, especially when focusing on small scales such as in this study. The error here is directly proportional to density, i.e. there will be 100% error in mass change if the assumed density is 100% different than it is in reality.*

We admit using wrong terms (especially in the section head) as well as unclear explanation in this section. Instead of a SMB, we rather want to show a simple underestimation of mass of East Antarctica here. Our calculation has been very simplistic, but regarding mass only we consider it to be principally correct.

We assumed an average thickness of 2000 m of firn and ice combined in East Antarctica. The firn column has a length of 93 m according to Herron and Langway (1980) with an initial density of 320 kg m$^{-3}$. This corresponds to 59 mWE. Calculating the water equivalent with our presented surface snow density, we get 61 mWE, meaning +3% mass in the firn column. This again corresponds to +0.11% in relation to the 2000 m firn and ice combined. This 0.11% of the water equivalent of East Antarctica (51.69 m SLE (Rignot et al., 2019)) leads to a ~5 cm SLE estimation.

We will probably add another section (and make section 4.3 more comprehensive) specifically discussing the applications of this dataset for satellite altimetry on ice sheets, which is more helpful than this mass budget calculation. For this purpose we will discuss our data more interdisciplinary with colleagues working with remote sensing data.

*Figure 11: How is surface roughness and sub-grid topography, e.g. using REMA, related to observed density in this figure? It is recommended to analyze the liners from Kohnen station from different seasons (as mentioned P9,L15/16) to show to what extent there is year-to-year variability.*

We also thought about this aspect in preparation for this manuscript and compared REMA surface and bedrock topography with our dataset. For drawing a significant conclusion from the comparison, we consider the local sampling distance (10 m) as too small and the regional distance (100 km) as too large. Still, this would make sense for adjusted sampling intervals (like 10 km or so) along transects with samples of representative density.

The resolution is simply a limiting factor here (in both dimensions, depth and space).

Your idea for samples on Kohnen station will be implemented (see your comment on P18, L18).

*P21, L5-8: First of all, a primary source of error in modelled snow density by the Ligtenberg et al. (2011) model could as well result from the meteorological driving data for the FDM simulations. Second, the text now seems to imply that more snow*

*redistribution leads to lower densities. However, it has been demonstrated that snow redistribution tends to increase*
185   *hardness/density (see Sommer at al. 2017, 2018).*

We will mention the meteorological forcing as a potential error source in our discussion.

Regarding the influence of snow redistribution on surface snow density, the wind speed and surface topography have to be taken into account. In wind speed maps (Parish, 1988; Sanz Rodrigo et al., 2012; van Lipzig et al., 2004), we see low wind vectors or mean wind speed from Kohnen station along the ice divide up the EAP and even lower values
190   for the region around Plateau Station. Rather than with only the mean wind speed, we also want to argue with the distribution of wind speed instead of average wind speed (generally low wind speeds on the East Antarctic Plateau with occasional strong winds causing drifting snow; the latter can happen more often at Kohnen station). We will back our discussion with the cited wind maps.

We are aware of the mentioned article (Sommer et al., 2018) and included it into the discussion as an example for
195   wind packing with (in contrast to the remote plateau) relatively high wind speed. Knowing that there are many complex possibilities for depositional processes, we want to contrast two scenarios:

1) At higher wind speeds, snow gets redistributed and sorted, after deposition that snow has a higher density mainly due to wind packing (sorted, high density, example by Sommer et al. (2018)).

2) Between dunes, soft snow (low density) gets deposited at low wind speeds, which causes a high variability (not
200   sorted, low density) over depth in a given period of time.

Scenario 2) is the explanation we give for the lower densities in the major part of the interior plateau, as we sum up in section 4.4.

*We thank the authors for taking the substantial time and effort to collect, describe, and distribute these density observations. That said, we believe these observations would best serve the community if they were also included in a unified and publicly*
205   *available dataset such as SUMup (Montgomery et. al., 2018).*

The data will be available on open-access repository Pangaea. We are happy to collaborate with our data on further datasets.

**2.2 Minor comments**

*P1, L11: Underestimations or overestimations.*

210   At the introductory part of the abstract, we kept the term 'uncertainties' deliberately as a general term. The under- or overestimations are dependent on the density measurements, of course. In our case, with higher density than assumed, we speak of an underestimation (s. same page, line 26)

*P1, L25: Density errors can be due to errors in parameterizations or atmospheric forcing.*

Will be added.

215 *P2, L3: "Greenland Ice Sheet"*

Will be corrected.

*P2, L3-5: Please reformulate. Either "accurate quantification is important" or "The current state and rate are some of the most important quantities…"*

Will be corrected as suggested.

220 *P2, L11: "Especially in the interior of the ice sheets, the exact surface snow density is a limiting factor in precision." Please amend why that is, with appropriate references if available.*

This sentence originated from a discussion with colleagues working with altimetry data. We found the following passage in Thomas et al. (2008) and will include it as reference: "Radar return-pulse waveforms from high-elevation parts of Antarctica are affected by various characteristics of the snowpack, such as snow density, distribution of ice,
225 wind-crust and depth-hoar layers (…), and by wind-induced surface roughness. (…) Near the coast, radar penetration into the snow is of far less concern than the local surface topography, which becomes quite rough, particularly in the most active parts of outlet glaciers where thinning rates are highest."

*P2, L23: "Small variability" → "A small part of variability" I assume.*

Will be corrected as suggested.

230 *P3, L1 and L9: "stratigraphic noise" please explain.*

Please see our answer to your major comment on P8, L19.

*P3, L3-6: "In this paper, we present surface snow density data with high precision from a traverse covering over 2000km on the East Antarctic Plateau (EAP). In order to avoid misunderstandings we follow Stenni et al. (2017) using the term EAP for the region higher than 2000m above sea level (asl). The coldest 10m firn temperature is recorded at Plateau Station (Picciotto*
235 *et al., 1971), which makes the area the best modern analog of glacial firn." Don't understand this section. Please explain in more detail.*

We do not have access to glacial-climate firn but firnification during glacial climate periods is modelled to calculate for example the Δage (the gas age-ice age difference), and to infer the phase relationship between temperature derived from the isotopes and the $CO_2$ concentration measured in ice cores. Modelling glacial-climate firn faces some
240 problems, e.g. at the pore close off (firn to ice transition). Firnification models simulate a deeper pore close off than

$\delta^{15}N$ data predict. In this sense we understand modern firn from the coldest regions of the EAP as the best modern analogue of glacial-climate firn (for some regions, e.g. Kohnen station). The acronym CoFi stands for Coldest Firn. Within the project, five 200 m firn cores have been drilled on the EAP to investigate the firn densification.

In general, we will move the project description to section 2.1 and add climatic information about the area. The snow profiles presented here were taken in the framework of this project. But as this information is not necessary for the further manuscript, we probably decide to remove this sentence.

*P3, L10 Average local snowpack density.*

Will be corrected.

*P4, Fig 1: Please add elevation contour labels.*

Will be added.

*P6, L20: The sentence "The trench surface was measured…" needs to be placed before P6, L19: "The total height difference between the lowest…"*

Will be corrected.

*P7, L9 weighted → weighed?*

Taken as suggested.

*P8, Fig 4: What explains the occasional large difference? Please add linear regression statistics.*

We add linear regression and elaborate in more detail in the text (see also comment on P8, L9-11).

*P8, L15: "Therefore, to quantify the 1m snowpack density we use L, to investigate smaller intervals we use the 1mµCT (Tab.1)" Since 1mµCT is CT density over 1m, and L the liner density over 1m, how should it be interpreted that 1mµCT can assist in investigating smaller intervals? Or should it read 0.1mµCT.*

The latter, it was a typo. Will be corrected.

*P8, L17: Snow density profiles?*

Will be replaced.

*P9, Section 2.6 Optical levelling needs to be placed before Section 2.2.3., since the optical levelling is already mentioned there.*

We carefully thought about this suggestion, and discuss a suitable solution.

The profiles investigated with optical levelling are completely independent from the liner sampling procedure and should be seen as an additional information, with which we want to address the topic of surface roughness. As the snow liners are the main samples in this study, we want to keep them as the first method presented.

270     Maybe we find an option to include the levelling in section 2.2.3.

*P9, L17: "Furthermore, a possible effect of the station itself should not be migrated into the other subsets." Please explain what effects are meant here.*

Around Kohnen station an effect of increased accumulation has been observed, which might influence density as well. We probably will erase this sentence, as the samples at Kohnen (previous studies) have an adequate distance to the

275     station itself.

*P10, Figure description: What is meant by raster?*

We started a common depth scale for the whole trench at the top of the highest (relative) snow profile. Then we calculated the mean value in 0.1 m intervals of each profile but according to the common depth scale (as visible in Fig. 5, bottom left). We used the term raster (or grid) as a more descriptive term, but will add a sentence like above

280     for clarity.

*P12, Table 2: Can you create maps of p_loc and sigma_1m?*

We will include a map in the next manuscript version.

*P18, L9: Please quantify dune height*

Dune heights on EAP can be up to 30-40 cm, we will include concrete values from section 3.5 here.

285     *P19, L4: What exactly leads you to make this claim. Could density errors be due to errors in atmospheric forcing? Temporal variability in snow density?*

As mentioned in your comment on P21, l5-8, atmospheric forcing was neglected in our manuscript as an error source and will be added to the discussion. Regarding temporal variability we refer to the comments above (P18, L18). Still, the large difference we observed between our dataset and Kaspers et al. (2004) & Ligtenberg et al. (2011) should

290     at least be partly accounted to the parameterization of surface snow density.

*Many figures have missing axes. Please correct.*

Missing axes will be added.

*References: please provide doi's for easy lookup of literature.*

We have added the DOIs to the references, but also want to mention, that the original template for EndNote did not
295    show DOIs by default. This should be updated by Copernicus.

Dallmayr, R., Freitag, J., Hörhold, M., Laepple, T., Lemburg, J., Della-Lunga, D., and Wilhelms, F.: A dual-tube sampling technique for snowpack studies, Journal of Glaciology, in review. doi: in review.

Fisher, D. A., Reeh, N., and Clausen, H. B.: Stratigraphic noise in time series derived from ice cores, Annals of Glaciology, 7,
300    76-83, doi: 10.3189/S0260305500005942, 1985.

Herron, M. M. and Langway, C. C.: Firn Densification - an Empirical-Model, Journal of Glaciology, 25, 373-385, doi: 10.3189/S0022143000015239, 1980.

Karlöf, L., Winebrenner, D. P., and Percival, D. B.: How representative is a time series derived from a firn core? A study at a low-accumulation site on the Antarctic plateau, Journal of Geophysical Research, 111, 11, doi: 10.1029/2006jf000552, 2006.

305    Kaspers, K. A., van de Wal, R. S. W., van den Broeke, M. R., Schwander, J., van Lipzig, N. P. M., and Brenninkmeijer, C. A. M.: Model calculations of the age of firn air across the Antarctic continent, Atmos. Chem. Phys., 4, 1365-1380, doi: 10.5194/acp-4-1365-2004, 2004.

Laepple, T., Hörhold, M., Münch, T., Freitag, J., Wegner, A., and Kipfstuhl, S.: Layering of surface snow and firn at Kohnen Station, Antarctica: Noise or seasonal signal?, Journal of Geophysical Research-Earth Surface, 121, 1849-1860, doi:
310    10.1002/2016jf003919, 2016.

Ligtenberg, S. R. M., Helsen, M. M., and van den Broeke, M. R.: An improved semi-empirical model for the densification of Antarctic firn, Cryosphere, 5, 809-819, doi: 10.5194/tc-5-809-2011, 2011.

Münch, T., Kipfstuhl, S., Freitag, J., Meyer, H., and Laepple, T.: Regional climate signal vs. local noise: a two-dimensional view of water isotopes in Antarctic firn at Kohnen Station, Dronning Maud Land, Climate of the Past, 12, 1565-1581, doi:
315    10.5194/cp-12-1565-2016, 2016.

Parish, T. R.: Surface winds over the Antarctic continent - a review, Reviews of Geophysics, 26, 169-180, doi: 10.1029/RG026i001p00169, 1988.

Rignot, E., Mouginot, J., Scheuchl, B., van den Broeke, M., van Wessem, M. J., and Morlighem, M.: Four decades of Antarctic Ice Sheet mass balance from 1979-2017, Proc. Natl. Acad. Sci. U. S. A., 116, 1095-1103, doi: 10.1073/pnas.1812883116,
320    2019.

Sanz Rodrigo, J., Buchlin, J.-M., van Beeck, J., Lenaerts, J. T. M., and van den Broeke, M. R.: Evaluation of the antarctic surface wind climate from ERA reanalyses and RACMO2/ANT simulations based on automatic weather stations, Climate Dynamics, 40, 353-376, doi: 10.1007/s00382-012-1396-y, 2012.

Sommer, C. G., Wever, N., Fierz, C., and Lehning, M.: Investigation of a wind-packing event in Queen Maud Land, Antarctica,
325    Cryosphere, 12, 2923-2939, doi: 10.5194/tc-12-2923-2018, 2018.

Thomas, R., Davis, C., Frederick, E., Krabill, W., Li, Y., Manizade, S., and Martin, C.: A comparison of Greenland ice-sheet volume changes derived from altimetry measurements, Journal of Glaciology, 54, 203-212, doi: 10.3189/002214308784886225, 2008.

van Lipzig, N. P. M., Turner, J., Colwell, S. R., and van Den Broeke, M. R.: The near-surface wind field over the Antarctic
330    continent, Int. J. Climatol., 24, 1973-1982, doi: 10.1002/joc.1090, 2004.

---

## Author Response (AR1)

**Interactive comments on**
**"Representative surface snow density on the East Antarctic Plateau" by Alexander H. Weinhart et al.**

5 *Comments by the referees will be displayed in italics,* the response from the authors in normal font.

**1 Review by Anonymous Referee #1**

*This paper describes density observations along two over-ice transects from Kohnen station to Dome F on the plateau of the East Antarctic ice sheet. The observational techniques are state-of-the-art, resulting in small errors and highly significant results. These results show that 0-1 m average density shows little variation along the traverse, with a mean value of about* 10 *355 kg m$^{-3}$. This is an important result, as it can be used to improve the snow/firn modules in (regional) climate models and the interpretation of satellite altimetry observations. However, the writing needs to be improved, as many formulations are unclear (for some examples, see below, but this listing is not exhaustive). The figure quality can also be improved in places.*

We thank the anonymous referee for his feedback on our manuscript. We carefully went through the manuscript again, clarified unclear passages and elaborated further on the influence of the presented data on satellite altimetry.

15 Generally we included axes in the figures where missing.

**1.1 Major comments**

*p. 1, l. 25: "The difference in the total mass equivalent of measured and modelled density yields a 3% underestimation by models, which translates into 5 cm sea level equivalent." It is unclear how these numbers are obtained, see comment below on Section 4.3.*

20 See comment on section 4.3.

*p. 2, l. 3: "Accurate quantification of the current state and rate of change of SMB is therefore one of the most important quantities..." A quantification is not a quantity. Please critically re-assess your formulations to improve this throughout the paper.*

Thank you very much for this advice. We reassessed unclear wording or passages in the text, for example the terms
25 quantification/quantity and precision/accuracy/representativeness. Necessary improvements were done especially in

section 2.4, where we added a more in-depth explanation of stratigraphic noise, and section 4.3 (see also comment on section 4.3).

*p. 3, l. 5: "The coldest 10 m firn temperature is recorded at Plateau Station (...), which makes the area the best modern analog of glacial firn." This is another example of a sentence that is really hard to understand. Coldest on Earth? What do you mean*
30 *by "an analogue of glacial firn"? Please clarify.*

We do not have access to glacial-climate firn but firnification during glacial climate periods is modelled to calculate for example the $\Delta$age (the gas age-ice age difference), and to infer the phase relationship between temperature derived from the isotopes and the $CO_2$ concentration measured in ice cores. Modelling glacial-climate firn faces some problems, e.g. at the pore close off (firn to ice transition). Firnification models simulate a deeper pore close off than
35 $\delta^{15}N$ data predict. In this sense we understand modern firn from the coldest regions of the EAP as the best modern analogue of glacial-climate firn (for some regions, e.g. Kohnen station). The acronym CoFi stands for Coldest Firn. Within the project, five 200 m firn cores have been drilled on the EAP to investigate the firn densification.
We moved the project description to section 2.1 and added climatic information about the area. The snow profiles presented here were taken in the framework of this project. But as this information about the project itself is not
40 necessary for the further manuscript, we decided to remove this sentence.

*Section 4.3: It is unclear to me how the density errors in previous studies lead to the SMB error results in a 5 cm sea level equivalent? Over what period? SMB is usually derived from regional climate models that quantify mass directly, i.e. irrespective of density. Mass changes by GRACE are also direct mass measurements, only satellite altimetry suffers from uncertainties in the density of the material at which the elevation change takes place, but this is valid for changes in elevation,*
45 *not for steady densities as presented here.*

This comment also refers to your general comment on a proper use of terms, here we refer to simply mass instead of a mass balance (4.6 in the revised manuscript). We want show the underestimation in mass in the firn column running the model by Herron and Langway (1980) with two different initial densities. According to our calculation, this underestimation (3% of the firn column) in East Antarctica is equivalent to a total mass of 5 cm sea level equivalent.
50 We updated the section with proper terms and described our line of thoughts at length. We also added another section (4.4 in the revised manuscript) with focus on the impact of our findings on satellite altimetry of ice sheets.

*Section 4.4: Are the Ligtenberg et al. (2011) data also valid for the first 1 m? I think they use a simple parametrization to calculate surface density, hardly a 'model' as it is called here. It would also be valuable to provide the time span covered by the first 1 m of snow, this will vary with accumulation. In how far can climate variability be responsible for part of the*
55 *differences with other studies? Based on its findings, does the current paper recommend to redefine 'surface density' as the average density of the first m? If so, this is an important recommendation that could be made more explicit.*

According to the dataset description, the data from Ligtenberg et al. (2011) are modelled (IMAU Firn Density Model) for the near-surface (0-1 m depth) and gridded to 33 km resolution. We added this information in the manuscript.

Also in preparation of this manuscript we stumbled upon several opinions about how 'surface snow density' is defined ('fresh snow density', 'near surface density' and 'density of the uppermost snow layer' are used sometimes arbitrary for different purposes). For practical reasons, 1 m intervals are not only a commonly used interval for density, but also stable water isotopes in surface snow (e.g. Masson-Delmotte et al., 2008). The suggestion to redefine the term surface snow density is an important point that was addressed. We elaborated a bit more on the advantages of 1 m surface snow density.

The problem of time span and the – in turn – advantage of 1 m density compared to smaller intervals, is tackled in the conclusion, but we emphasised in the discussion a bit more. As mentioned before, we added a section about the climatic setting of the area. This includes information about accumulation rates as well as temperature. At Kohnen station the accumulation rate is 64 mm we a$^{-1}$ (we = water equivalent) (Oerter et al., 1999) with increasing tendency over the last decades (Medley et al., 2018), at Dome Fuji 27.3 mm we a$^{-1}$ has been measured (Hoshina et al., 2016). For the locations along the traverse a precise value is difficult to obtain. A 1 m snow profile therefore can cover a time period of four years at Kohnen station to 20 years on the remote Plateau.

Regarding your comment on climate variability: if we understood correctly, you ask whether changing temperature can be responsible for a difference between two density datasets. Here we want to refer to section 4.3 (revised manuscript), in which we elaborate on this thought. We show, that climatic driven changes in density are too low to ascribe the difference in density between the datasets only to temperature.

**1.2 Minor comments**

*p. 1, l. 10: "Wrong estimates of snow and firn density can lead to significant underestimations of the surface mass balance." underestimations of → uncertainties in*

*p. 1, l. 17: "liner"? This has not been explained yet, so please don't use it here.*

*p. 1, l. 23: "provided by a regional climate model" These models usually don't 'provide' density, but either prescribe it or use a simplified expression based on temperature, wind etc. Suggest to replace by 'used'.*

*p. 1, l. 25 and further: Note that regional climate models DO explicitly calculate accumulated mass, so using a wrong surface density does not influence the surface mass balance directly, only indirectly (through blowing snow threshold friction velocity, vertical heat transport in snow affecting surface temperature and hence sublimation etc.).*

*p. 2, l. 2: on → of; Greenlandic → Greenland.*

*p. 2, l. 5: "Satellite altimetry is state of the art" → Satellite altimetry is a state of the art method/technique...*

*p. 2, l. 13: "snow density is parameterized" → snow density in models is often parameterized*

*p. 2, l. 30: This sentence is unclear, please reformulate.*

*p. 3, l. 4: Remove "In order to avoid misunderstandings"*

90 *p. 3, l. 5: coldest/warmest temperatures → lowest/highest temperatures (change this throughout the text, please)*

  *p. 9, l. 8: good → well*

  *p. 18, l. 23: This AWS has been installed and serviced by Utrecht University and AWI; please provide proper credit.*

  *Figures: Please include solid axes where missing.*

   All minor comments were taken as suggested by the referee.

95

**2 Review by Eric Keenan, Nander Wever, Jan Lenaerts**

*The authors present a suite of highly accurate surface density measurements taken during a traverse in Dronning Maud Land, East Antarctica. These observations have the potential to offer the ice sheet scientific community a unique and very useful dataset to evaluate and improve snow and firn densification models. The authors present principally interesting spatial analysis of the measured snow density, which adds to the presented data. That said, we have significant concerns that should be addressed before publication, namely*

*1) a more detailed description of density uncertainty quantification,*

*2) the method used to quantify the impact of density on surface mass balance retrieval, and*

*3) a more detailed description of observed small scale density variability in the top 1 m presented in Figure 5, as well as its potential drivers and implications for interpretation of satellite altimetry observations.*

*Please find a more detailed description of these suggestions and others broken into major and minor comments below.*

> We kindly thank Eric Keenan, Nander Wever and Jan Lenaerts for their detailed and productive feedback. The deliberate comments definitely helped to improve this manuscript, in particular the discussion and application of the presented dataset.
>
> Regarding the impact on satellite altimetry, we added another section in the manuscript.

**2.1 Major comments**

*Section 2.1: This section would benefit from a general discussion of weather and climate conditions in the area, in relation to how they may impact surface density (in terms of variability of yearly accumulation rates, wind speed and temperature). This would help setting the stage for the discussion in section 4.4.*

> We agree, rearranged section 2.1 and added a climatic overview about the area (especially regarding temperature and accumulation rate). This should also clarify unclear passages in the text (see also minor comment on P3, L3-6).

*P7, L12: "Breaks and lost snow in the snow profiles haven been corrected." This needs more explanation.*

> s. comment on P8, L15

*P8, L9-11: "It is generally possible that at the liner top and bottom some snow is lost, but as the exact snow volume is determined with the μCT, we overcome this error source." It's not clear how the microCT can compensate for errors due to lost snow. It's the same liner that's measured by microCT and the scale, so if the snow is lost, both methods should be affected.*

> s. comment on P8, L15

*P8, L15: "Therefore…" I don't see how this sentence follows logically from the previous section.*

125     We realized that parts of section 2.3 were not coherent enough and updated it. Now the logical sequence as well as the description of uncertainty in snow density should be clearer.

Regarding the missing snow: For the calculation of the µCT density only the central segment of the liner is used as scattering effects at the outer parts of the liner occur. The used segment corresponds to less than half of the snow volume in the liner. Missing snow at the edges of the profile does not influence the µCT scan.

130     It is generally possible that the snow profiles are subject to compression during sampling or transport. Therefore the exact snow volume determined with the µCT is rescaled to the original 1 m length (length of every single snow profile is determined individually) to avoid this potential error source. But lost snow in the liner (or at top or bottom) e.g. in non-cohesive layers (such as depth hoar layers) can lead to lower densities for the volumetrically calculated snow density.

135     Still, for 1 m segments, the volumetrically derived density has a higher precision as Fig. 7 shows; we added more details the text.

*P8, L19: "spatially independent" Not really clear. Is the measurement setup in Fig.3 considered spatially independent? I.e., are liners X, A, B, C considered spatially independent? According to the text they are, but those four liners are not really independent.*

140     Fisher et al. (1985) defined local noise as "random element caused by the surface irregularities", which is present in any taken snow profile or ice core. Stratigraphic noise and depositional noise are used as equivalent terms for local noise as they are more descriptive. This noise is mainly caused by spatially inhomogeneous deposition in combination with wind, leading to snow patches or dune structures that usually have a spatial extent of several meters.

To still be able to get a representative value or profile (of density or other parameters) at a given spot, several samples

145     have to be taken at a distance, at which they have not recorded the same depositional (or stratigraphic) noise. For example, samples should not be taken from the same dune or snow patch. By stacking or averaging the samples, the noise is minimized. The (minimum) sampling distance between two samples was quantified by Laepple et al. (2016) with 5-10 m, as the correlation factor between single profiles decreases with increasing distance. Also the sampling distance in this study was chosen according to this finding. In an (unpublished) test using our density profiles we also

150     come to a similar result. Note: Laepple et al. (2016) sampled perpendicular to the dune direction like we did in the OIR trench. For sampling parallel to the wind, the sampling distance between two samples should be higher.

This condition (not recording the same depositional noise several times) was called 'spatially independent' in our manuscript. The problem of stratigraphic noise is generally higher in regions with low accumulation and has also been recorded for e.g. isotopes (Karlöf et al., 2006; Münch et al., 2016).

155     For a better understanding, we added passages of the explanation above in the text.

*Section 2.4: This section is difficult to comprehend, and is written very compact. Particularly, please expand on: "This way we use the maximum sample size without an artificially caused bias in the data."*

For the whole section, there might be many different approaches to determine a certain number of precision for our data. The method we use has also recently been applied in a study by Dallmayr et al. (in review). We are aware of the not-straightforward method and tried to explain in more detail. With 'artificially caused bias' we mean the instance of arbitrary picking sets of different numbers of $\rho_L$ (e.g. a certain number of independent profiles out of the 30 trench profiles). Instead we suggest to take the maximum possible number from the beginning. For more clarity, we also put the formula

$$\sigma_n = \sigma_H^{1m} \Big/ \sqrt{[2; n\text{-}1]}$$

in a more explicit position in the manuscript and changed x to [2;n-1], which makes is more understandable.

*Figure 5: The large variability in observed density, particularly in the top meter, is very interesting and is a very nice inclusion in this paper. Can you please elaborate on what might cause this (surface topography? winds?) and what this variability means for interpretation of satellite altimetry. In particular, the observed variability is in apparent contrast with the title of this paper "Representative surface snow density...". If surface density is highly variable, is a representative density truly the best approach or should the scientific community make an effort to model this variability? A related comment is that Section 3.5 is really short and only mentions results. It's not clear which conclusions the authors draw from this and how it is important to understand surface density variability.*

The surface topography is definitely one of the driving factors for the high horizontal variability of density we see in the top 20 cm in the OIR trench. It can be seen as a complex interaction between accumulation (a combination of calm diamond dust deposition as well as event-based accumulation), redistribution of soft snow by moderate wind and the existing topography, with the topography being the dominant factor (height & shape). Additionally, long residence time of snow at the surface due to low accumulation rates enhances the chance of metamorphism and sublimation (due to the vertical temperature gradients and low humidity), which also can have an impact on the surface snow density. We expanded the discussion of the driving factors in section 4.1.

Indeed, the high (local) variability at the snow surface (especially in the upper 20 cm) is a major finding of this manuscript. But as a consequence this is also the reason, why we argue for the use of a 1 m mean density as a more robust parameter for surface snow density. This way the density variability at the uppermost surface is compensated by using enough depth (or "annual layers", in other terms) without having the influence of a densification effect. We emphasized this statement in the discussion and the conclusion.

185       Otherwise, the high variability at the surface can also be an argument for a representative density obtained with the method we presented. Especially as altimetry measurements have a certain footprint area, representative regional density values can be of particular interest.

      Despite the argument for the 1 m snow density, from our point of view a representative vertical variability is a more important aspect than modelling the horizontal variability. This is why we included the density distribution of density (Figure 9) in the manuscript and calculated the distribution also according to the presented subareas. The vertical
190 variability can be interpreted as a measure of snowpack layering (i.e. layers of high and low density). The layering definitely has a strong influence on the densification from snow to ice on one hand (and therefore also on the bubble close-off at the firn-ice transition and subsequent effects). On the other hand, also satellite altimetry can use this information of vertical layering due to penetration depth into the snow and reflection at layer boundaries, but the topic
195 of snowpack layering demands another dimension (esp. further parameters to describe the snowpack) that goes beyond the current manuscript frame.

      Regarding section 3.5: We included this – indeed – very short section as a baseline for the argument of dune height and surface topography on the East Antarctic Plateau. We mention the relation of dune height to accumulation rate later in the manuscript. One might argue, that the dune height of 30-40 cm is not that high. But in contrast to the
200 annual accumulation of ~10 cm, the dune height becomes enormous. We marked in the text when we refer to the measured surface heights.

*P11, L1-3 and in the following sections: If the standard deviation for 1m sampling intervals is 5-10 kg/m3, how can the error quantification for the average be only +/- 2 kg/m3?*

      We assume, you refer to the horizontal standard deviation, as also displayed in Fig. 5.
205       The horizontal standard deviation ($\sigma^{1m}_H$) for the liner mean density ($\rho_L$) in the OIR trench is 9.63 kg m$^{-3}$. The standard error then is defined as the standard deviation divided by the square root of the samples size: ($\sigma^{1m}_H / \sqrt{30}$) = 1.76 kg m$^{-3}$ ≈ 2 kg m$^{-3}$

*Figure 8: Observations report a near uniform mean density in the four different subregions. For me, this is a surprising result. How might the different dates the observations were taken affect the measured density, i.e. do you expect a seasonal cycle in*
210 *surface density. The way this dataset is currently presented, does not take into account this possibility. Additionally, if you are not already planning on doing so, can you please include exact observations date and time in the final dataset publication on Pangaea?*

      To be honest, we also expected to see a larger difference between the subregions (in both, mean density (Fig. 8) and density distribution (Fig. 9)) and a clearer trend along the traverse before the measurements.
215       To clarify your remark we added the sampling date at each location in table 2 (we planned to add the sampling date to the Pangaea dataset). From first (14.12.2016) to last (28.01.2017) sampling date we do not expect to see a seasonal

cycle or bias due to sampling, especially as we did not notice significant accumulation during the traverse (main synoptic features: some diamond dust above 3500 mSL during the nights, during some days moving/drifting snow. The temperatures varied between -20 and -40°C during the night). We also added a sentence on that to section 4.4.

220 From extensive sampling programs at Kohnen station we can generally say, that it is possible to detect seasonal cycles in density profiles – but only with a sufficient amount of samples (if local noise can be eliminated) (Laepple et al., 2016). On the EAP, this may be more difficult as the accumulation rate is much lower. To derive a representative density profile from the OIR trench can be a part of a future study.

*P18, L18: This line mentions natural variability due to antecedent weather conditions. Section 4.2 needs to put the analysis*
225 *based on climatological trends in perspective to possible year-to-year variability due to antecedent weather conditions. Since accumulation depths in dunes could be up to 30cm, this may impact top 1m density significantly.*

In this context, high mean density of single profiles can also be explained with a high percentage of dune snow in a profile. A major problem here is that we do not know anything about the persistence of the surface roughness or surface features. If the surface is reshaped once or twice a year by stronger wind (>10 ms$^{-1}$), it is hard to attribute this
230 variability to a variability of climate. Also the distribution of snow accumulation during the year is poorly known (keeping in mind the differences at Kohnen station with higher synoptic influence than the remote plateau with diamond dust deposition).

We elaborated on the year-to-year variability a bit further by comparing samples from Kohnen station from different seasons (16/17 and 18/19). Please note here, that the temporal and spatial variations cannot be decoupled completely,
235 as we cannot sample the exact position again. But as the samples are taken as close as possible to the original sampling positions (several cm), we consider the spatial variation to be very low.

For further discussion about the climatic influence on surface snow density, we want to refer to section 4.3. (revised manuscript), where we test, whether the discrepancy in density between two datasets can be ascribed to rising temperatures in DML. We come to the conclusion, that the warming of 1°C alone cannot explain the difference in
240 1 m surface snow density and ascribe this to the stratigraphic noise.

*Section 4.3: The authors aim to provide the impact of density on the uncertainty in SMB, but they fail to do this correctly. First of all, 3% uncertainty in the firn column does not directly translate to a 3% uncertainty in the overall firn+ice column (since there is much more ice than firn on East Antarctica). Secondly, this calculation pertains to mass, not mass balance (i.e. the change in mass per unit time). Instead, the authors should think about focusing on the application to altimetry, which needs*
245 *surface density to convert volume to mass. As most of the elevation changes on East Antarctica measured by altimetry are SMB-driven, the observed elevation change will be associated to the layer of recently accumulated/ablated snow/firn, with an extremely spatially and temporally variable density. Since this (near-)surface density is much more variable than at 1 m, this volume-to-mass conversion is highly uncertain, especially when focusing on small scales such as in this study. The error here*

*is directly proportional to density, i.e. there will be 100% error in mass change if the assumed density is 100% different than*
250 *it is in reality.*

We admit using wrong terms (especially in the section head) as well as unclear explanation in this section. Instead of a SMB, we rather want to show a simple underestimation of mass of East Antarctica here. Our calculation has been very simplistic, but regarding mass only we consider it to be principally correct.

We assumed an average thickness of 2000 m of firn and ice combined in East Antarctica. The firn column has a length
255 of 93 m according to Herron and Langway (1980) with an initial density of 320 kg m$^{-3}$. This corresponds to 59 mWE. Calculating the water equivalent with our presented surface snow density, we get 61 mWE, meaning +3% mass for the same depth as above. This again corresponds to +0.11% in relation to the 2000 m of firn and ice combined. This 0.11% of the water equivalent of East Antarctica (51.69 m SLE (Rignot et al., 2019)) leads to a ~5 cm SLE estimation. We made section 4.6 (revised manuscript) more comprehensive and added another section specifically discussing the
260 applications of this dataset for satellite altimetry on ice sheets.

*Figure 11: How is surface roughness and sub-grid topography, e.g. using REMA, related to observed density in this figure? It is recommended to analyze the liners from Kohnen station from different seasons (as mentioned P9,L15/16) to show to what extent there is year-to-year variability.*

We also thought about this aspect in preparation for this manuscript and compared REMA surface and bedrock
265 topography with our dataset. The resolution is simply a limiting factor here (in both dimensions, depth and space). For drawing a significant conclusion from the comparison, we consider the local sampling distance (10 m) as too small and the regional distance (100 km) as too large. Still, this would make sense for adjusted sampling intervals (like 10 km or so) along transects with samples of representative density.

Your idea for samples on Kohnen station has been implemented (see your comment on P18, L18).

270 *P21, L5-8: First of all, a primary source of error in modelled snow density by the Ligtenberg et al. (2011) model could as well result from the meteorological driving data for the FDM simulations. Second, the text now seems to imply that more snow redistribution leads to lower densities. However, it has been demonstrated that snow redistribution tends to increase hardness/density (see Sommer at al. 2017, 2018).*

We mentioned the meteorological forcing as a potential error source in our discussion.
275 Regarding the influence of snow redistribution on surface snow density, the wind speed and surface topography have to be taken into account. In wind speed maps (Parish, 1988; Sanz Rodrigo et al., 2012; van Lipzig et al., 2004), we see low wind vectors or mean wind speed from Kohnen station along the ice divide up the EAP and even lower values for the region around Plateau Station. Rather than with only the mean wind speed, we also want to argue with the distribution of wind speed instead of average wind speed (generally low wind speeds on the East Antarctic Plateau

280 with occasional strong winds causing drifting snow; the latter can happen more often at Kohnen station). We backed our discussion with the cited wind maps.

We are aware of the mentioned article (Sommer et al., 2018) and included it into the discussion as an example for wind packing with (in contrast to the remote plateau) relatively high wind speed. Knowing that there are many complex possibilities for depositional processes, we want to contrast two scenarios:

285
1) At higher wind speeds, snow gets redistributed and sorted, after deposition that snow has a higher density mainly due to wind packing (sorted, high density, example by Sommer et al. (2018)).

2) Soft snow (low density) gets deposited at low wind speeds, which causes a high variability (not sorted, low density) over depth in a given period of time.

Scenario 2) is the explanation we give for the lower densities in the major part of the interior plateau, as we sum up
290 in section 4.4.

*We thank the authors for taking the substantial time and effort to collect, describe, and distribute these density observations. That said, we believe these observations would best serve the community if they were also included in a unified and publicly available dataset such as SUMup (Montgomery et. al., 2018).*

The data will be available on open-access repository Pangaea. We are happy to collaborate with our data on further
295 datasets.

**2.2 Minor comments**

*P1, L11: Underestimations or overestimations.*

At the introductory part of the abstract, we kept the term 'uncertainties' deliberately as a general term. The under- or overestimations are dependent on the density measurements, of course. In our case, with higher density than assumed,
300 we speak of an underestimation (s. same page, line 26).

*P1, L25: Density errors can be due to errors in parameterizations or atmospheric forcing.*

Added.

*P2, L3: "Greenland Ice Sheet"*

Corrected.

305 *P2, L3-5: Please reformulate. Either "accurate quantification is important" or "The current state and rate are some of the most important quantities…"*

Corrected as suggested.

*P2, L11: "Especially in the interior of the ice sheets, the exact surface snow density is a limiting factor in precision." Please amend why that is, with appropriate references if available.*

310      This sentence originated from a discussion with colleagues working with altimetry data. We found the following passage in Thomas et al. (2008) and included it as reference: "Radar return-pulse waveforms from high-elevation parts of Antarctica are affected by various characteristics of the snowpack, such as snow density, distribution of ice, wind-crust and depth-hoar layers (…), and by wind-induced surface roughness. (…) Near the coast, radar penetration into the snow is of far less concern than the local surface topography, which becomes quite rough, particularly in the

315      most active parts of outlet glaciers where thinning rates are highest."

*P2, L23: "Small variability" → "A small part of variability" I assume.*

     Corrected as suggested.

*P3, L1 and L9: "stratigraphic noise" please explain.*

     Please see our answer to your major comment on P8, L19.

320 *P3, L3-6: "In this paper, we present surface snow density data with high precision from a traverse covering over 2000km on the East Antarctic Plateau (EAP). In order to avoid misunderstandings we follow Stenni et al. (2017) using the term EAP for the region higher than 2000m above sea level (asl). The coldest 10m firn temperature is recorded at Plateau Station (Picciotto et al., 1971), which makes the area the best modern analog of glacial firn." Don't understand this section. Please explain in more detail.*

325      We do not have access to glacial-climate firn but firnification during glacial climate periods is modelled to calculate for example the $\Delta$age (the gas age-ice age difference), and to infer the phase relationship between temperature derived from the isotopes and the $CO_2$ concentration measured in ice cores. Modelling glacial-climate firn faces some problems, e.g. at the pore close off (firn to ice transition). Firnification models simulate a deeper pore close off than $\delta^{15}N$ data predict. In this sense we understand modern firn from the coldest regions of the EAP as the best modern

330      analogue of glacial-climate firn (for some regions, e.g. Kohnen station). The acronym CoFi stands for Coldest Firn. Within the project, five 200 m firn cores have been drilled on the EAP to investigate the firn densification.
     We moved the project description to section 2.1 and added climatic information about the area. The snow profiles presented here were taken in the framework of this project. But as this information about the project itself is not necessary for the further manuscript, we decided to remove this sentence.

335 *P3, L10 Average local snowpack density.*

     Corrected.

*P4, Fig 1: Please add elevation contour labels.*

Added.

*P6, L20: The sentence "The trench surface was measured…" needs to be placed before P6, L19: "The total height difference between the lowest…"*

Corrected.

*P7, L9 weighted → weighed?*

Taken as suggested.

*P8, Fig 4: What explains the occasional large difference? Please add linear regression statistics.*

We added linear regression and elaborated on the snow density uncertainty in more detail in the text (see also comment on P8, L9-11).

*P8, L15: "Therefore, to quantify the 1m snowpack density we use L, to investigate smaller intervals we use the 1mµCT (Tab.1)" Since 1mµCT is CT density over 1m, and L the liner density over 1m, how should it be interpreted that 1mµCT can assist in investigating smaller intervals? Or should it read 0.1mµCT.*

The latter, it was a typo. Corrected.

*P8, L17: Snow density profiles?*

Replaced.

*P9, Section 2.6 Optical levelling needs to be placed before Section 2.2.3., since the optical levelling is already mentioned there.*

We carefully thought about this suggestion, and discussed a suitable solution.
The profiles investigated with optical levelling are completely independent from the liner sampling procedure and should be seen as an additional information, with which we want to address the topic of surface roughness. As the snow liners are the main samples in this study, we want to keep them as the first method presented.
We moved the levelling of the OIR trench into section 2.6.

*P9, L17: "Furthermore, a possible effect of the station itself should not be migrated into the other subsets." Please explain what effects are meant here.*

Around Kohnen station an effect of increased accumulation has been observed, which might influence density as well.

We erased this sentence, as the samples at Kohnen (previous studies) have an adequate distance to the station itself.

*P10, Figure description: What is meant by raster?*

365    We started a common depth scale for the whole trench at the top of the highest (relative) snow profile. Then we calculated the mean value in 0.1 m intervals of each profile but according to the common depth scale (as visible in Fig. 5, bottom left). We used the term raster (or grid) as a more descriptive term, but added a sentence like above for clarity.

*P12, Table 2: Can you create maps of p_loc and sigma_1m?*

370    We included a map in the appendix of the manuscript.

*P18, L9: Please quantify dune height*

Dune heights on EAP can be up to 30-40 cm, we included concrete values from section 3.5 here.

*P19, L4: What exactly leads you to make this claim. Could density errors be due to errors in atmospheric forcing? Temporal variability in snow density?*

375    As mentioned in your comment on P21, l5-8, atmospheric forcing was neglected in our manuscript as an error source and was added to the discussion, without being able to determine the exact contribution.

The large difference we observed between our dataset and Kaspers et al. (2004) & Ligtenberg et al. (2011) should at least be partly accounted to the parameterization of surface snow density.

Regarding temporal variability we refer to the comments above (P18, L18).

380    *Many figures have missing axes. Please correct.*

Missing axes added.

*References: please provide doi's for easy lookup of literature.*

We have added the DOIs to the references, but also want to mention, that the original template for EndNote did not show DOIs by default. This should be updated by Copernicus.

385

390    Dallmayr, R., Freitag, J., Hörhold, M., Laepple, T., Lemburg, J., Della-Lunga, D., and Wilhelms, F.: A dual-tube sampling technique for snowpack studies, Journal of Glaciology, in review. doi: in review.

Fisher, D. A., Reeh, N., and Clausen, H. B.: Stratigraphic noise in time series derived from ice cores, Annals of Glaciology, 7, 76-83, doi: 10.3189/S0260305500005942, 1985.

Herron, M. M. and Langway, C. C.: Firn Densification - an Empirical-Model, Journal of Glaciology, 25, 373-385, doi: 10.3189/S0022143000015239, 1980.

Hoshina, Y., Fujita, K., Iizuka, Y., and Motoyama, H.: Inconsistent relationships between major ions and water stable isotopes in Antarctic snow under
395    different accumulation environments, Polar Science, 10, 1-10, doi: 10.1016/j.polar.2015.12.003, 2016.

Karlöf, L., Winebrenner, D. P., and Percival, D. B.: How representative is a time series derived from a firn core? A study at a low-accumulation site on the Antarctic plateau, Journal of Geophysical Research, 111, 11, doi: 10.1029/2006jf000552, 2006.

Kaspers, K. A., van de Wal, R. S. W., van den Broeke, M. R., Schwander, J., van Lipzig, N. P. M., and Brenninkmeijer, C. A. M.: Model calculations of the age of firn air across the Antarctic continent, Atmos. Chem. Phys., 4, 1365-1380, doi: 10.5194/acp-4-1365-2004, 2004.

400    Laepple, T., Hörhold, M., Münch, T., Freitag, J., Wegner, A., and Kipfstuhl, S.: Layering of surface snow and firn at Kohnen Station, Antarctica: Noise or seasonal signal?, Journal of Geophysical Research-Earth Surface, 121, 1849-1860, doi: 10.1002/2016jf003919, 2016.

Ligtenberg, S. R. M., Helsen, M. M., and van den Broeke, M. R.: An improved semi-empirical model for the densification of Antarctic firn, Cryosphere, 5, 809-819, doi: 10.5194/tc-5-809-2011, 2011.

Masson-Delmotte, V., Hou, S., Ekaykin, A., Jouzel, J., Aristarain, A., Bernardo, R. T., Bromwich, D., Cattani, O., Delmotte, M., Falourd, S., Frezzotti, M.,
405    Gallee, H., Genoni, L., Isaksson, E., Landais, A., Helsen, M. M., Hoffmann, G., Lopez, J., Morgan, V., Motoyama, H., Noone, D., Oerter, H., Petit, J. R., Royer, A., Uemura, R., Schmidt, G. A., Schlosser, E., Simoes, J. C., Steig, E. J., Stenni, B., Stievenard, M., van den Broeke, M. R., de Wal, R. S. W. V., de Berg, W. J. V., Vimeux, F., and White, J. W. C.: A review of Antarctic surface snow isotopic composition: Observations, atmospheric circulation, and isotopic modeling, Journal of Climate, 21, 3359-3387, doi: 10.1175/2007jcli2139.1, 2008.

[revised manuscript text omitted]

---

## Author Response (AR2)

**Review report on**
**"Representative surface snow density on the East Antarctic Plateau" by Alexander H. Weinhart et al.**

*Comments by the referees will be displayed in italics,* the response from the authors in normal font

**1 Review by Eric Keenan, Nander Wever, Jan Lenaerts**

*The authors improved the manuscript considerably and took all comments into consideration. We think that the manuscript can be accepted after taking some minor issues, that we point out below, into consideration. Page and line numbers refer to the track-changed version in "tc-2020-14-author_response-version2.pdf"*

*Introduction L14-19: This section needs some more additional information. It states that new snow density is validated with field measurements (L15), but that field measurements show that the model underestimates snow density (L16-17). So probably it needs to be made more precise how the field measurements differ, or if it's a difference in definition of new/surface snow density.*

A big problem we face here, as also stated in our first reply, is an unclear definition of 'surface snow density', 'fresh snow density' (equivalent to 'new snow density') and the 'snow density of the uppermost layer'.

For our understanding, the latter of the three is the vaguest one, as often it is not clear, how a snow layer is defined and a distinct stratification is not visible in most profiles on the East Antarctic Plateau (therefore also not used in the manuscript). We understand the 'surface snow density' as the density from the surface to 1 m depth (like we state in the conclusions) and the term 'fresh snow density' as the uppermost part of a snowpack, in particular the upper 5 cm or 10 cm. Considering these definitions, we updated the passage.

*P3, L16: "joint venture of" to "joint venture between"*

Changed.

*P5, L1/2: Are both of these values air or firn temperatures?*

In the old manuscript, we referred to the annual mean air temperature at Kohnen Station and the 10 m firn temperature at Plateau Station. We inserted the 10 m firn temperature at Kohnen Station (-44.5°C according to Oerter et al. (2000)) for better comparability.

*P5, L4: Please specify the period over which the accumulation rate was determined ("used to be").*

According to Oerter et al. (2004) for the period 1200 to 2000. Added.

*P10, L17: It's still somewhat challenging to fully comprehend the manuscript at this point. Please explicitly write down how you arrived at 7 as the maximum possible number with spatially independent samples. I assume it must have to do with the earlier declared independent distance of 5m (L12), and 30 profiles over 50m, as stated earlier.*

The assumption is correct. We tried to rephrase the according sentences for clarity.

*P25, L29/30: "We can reduce the sampling error from up to ±4% (Conger and McClung, 2009) by the liner technique (this study and e.g. Schaller et al., 2016) to less than 2% relative error for a 1 m snow density." After revision, this sentence is now confusing, as it sounds like the liner technique has a +/- 4% error. I assume it should read something like: "by using the liner technique (this study and e.g. Schaller et al., 2016), we can reduce the sampling error from up to ±4% for other measurement techniques (Conger and McClung, 2009) to less than 2% relative error for a 1 m snow density."*

We took your suggested sentence as it is.

*Table 2: Change "Dez" to "Dec".*

Changed.

*Equations 2 and 3: Do those correspond to 1 m density? Or new snow density? If they apply to new snow density, is this a fair analysis?*

According to Kaspers et al. (2004) and Sugiyama et al. (2012), both parameterizations refer to 'surface snow density' (we did not find any depth interval in the definition in Kaspers et al. (2004)).
Assuming a similar understanding as we pointed out in the answer to your first comment, this comparison should be fair. But even if a smaller interval is meant in the parameterizations: the 1 m density and the 10 cm density do not show a systematic difference (s. Figure 10 in our manuscript). The snow density in the uppermost part is not generally smaller, sometimes the density is even higher than the 1 m mean. Therefore, also for this case, we assume the comparison to be fair.
In the final version, we will use the term surface snow density parameterization.

*P22, L7: Is the resolution not 27 km? (see Ligtenberg et al, 2011)*

Corrected.

*Section 4.6: This section is still not very clear. I think this section should either be dropped completely or a detailed description*

55 *(equations and all logic explicitly written out) should be added. The authors write: "As the firn-ice-transition is not as deep at the coast as on the EAP, we consider this calculation to be somewhat overestimated." But it is not shown in the paper that the same surface density bias exists at the coast. In the second to last sentence in the conclusion, it is not clear to which mass "mass" (P26,L33) is referring to. Is it "surface mass balance"?*

We agree, that for an understandable mass budget of East Antarctica some more assumptions have to be taken into

60 account and will drop the transfer of mass to sea level equivalent completely. Also a referring sentence in the conclusion was deleted.

But we consider the information of an underestimated firn depth when using a modelled surface snow density, which is considerably lower than the field measurements, for interesting and necessary in several aspects. Therefore, we changed the title of the chapter and added some words for clarity.

65 In the mentioned sentence in the conclusion, we refer to SMB. Changed.

[revised manuscript text omitted]